# Unsupervised Learning under Latent Label Shift

**Manley Roberts***     **Pranav Mani***     **Saurabh Garg**     **Zachary C. Lipton**
Carnegie Mellon University
{manleyroberts,zlipton}@cmu.edu; {pmani, sgarg2}@cs.cmu.edu

## Abstract

What sorts of structure might enable a learner to discover classes from unlabeled data? Traditional approaches rely on feature-space similarity and heroic assumptions on the data. In this paper, we introduce unsupervised learning under *Latent Label Shift* (LLS), where we have access to unlabeled data from multiple domains such that the label marginals $p_d(y)$ can shift across domains but the class conditionals $p(\mathbf{x}|y)$ do not. This work instantiates a new principle for identifying classes: elements that shift together group together. For finite input spaces, we establish an isomorphism between LLS and topic modeling: inputs correspond to words, domains to documents, and labels to topics. Addressing continuous data, we prove that when each label's support contains a separable region, analogous to an anchor word, oracle access to $p(d|\mathbf{x})$ suffices to identify $p_d(y)$ and $p_d(y|\mathbf{x})$ up to permutation. Thus motivated, we introduce a practical algorithm that leverages domain-discriminative models as follows: (i) push examples through domain discriminator $p(d|\mathbf{x})$; (ii) discretize the data by clustering examples in $p(d|\mathbf{x})$ space; (iii) perform non-negative matrix factorization on the discrete data; (iv) combine the recovered $p(y|d)$ with the discriminator outputs $p(d|\mathbf{x})$ to compute $p_d(y|x)\ \forall d$. With semi-synthetic experiments, we show that our algorithm can leverage domain information to improve upon competitive unsupervised classification methods. We reveal a failure mode of standard unsupervised classification methods when data-space similarity does not indicate true groupings, and show empirically that our method better handles this case. Our results establish a deep connection between distribution shift and topic modeling, opening promising lines for future work[2].

## 1 Introduction

Discovering systems of categories from unlabeled data is a fundamental but ill-posed challenge in machine learning. Typical unsupervised learning methods group instances together based on feature-space similarity. Accordingly, given a collection of photographs of animals, a practitioner might hope that, in some appropriate feature space, images of animals of the same species should be somehow similar to each other. But why should we expect a clustering algorithm to recognize that dogs viewed in sunlight and dogs viewed at night belong to the same category? Why should we expect that butterflies and caterpillars should lie close together in feature space?

In this paper, we offer an alternative principle according to which we might identify a set of classes: we exploit distribution shift across times and locations to reveal otherwise unrecognizable groupings among examples. For example, if we noticed that whenever we found ourselves in a location where butterflies are abundant, caterpillars were similarly abundant, and that whenever butterflies were scarce, caterpillars had a similar drop in prevalence, we might conclude that the two were tied to the same underlying concept, no matter how different they appear in feature space. In short, our principle suggests that latent classes might be uncovered whenever *instances that shift together group together*.

---

*Equal contribution.

[2] Code is available at https://github.com/acmi-lab/Latent-Label-Shift-DDFA.

36th Conference on Neural Information Processing Systems (NeurIPS 2022).

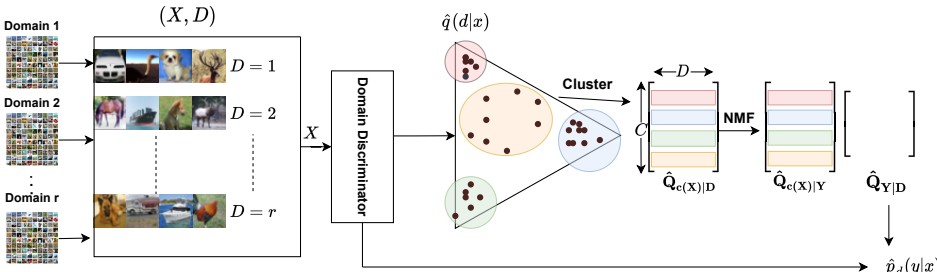

Figure 1: **Schematic of our DDFA algorithm**. After training a domain discriminator, we (i) push all data through the discriminator; (ii) cluster the data based on discriminator outputs; (iii) solve the resulting discrete topic modeling problem and then combine $\widehat{q}(d|x)$ and $\widehat{q}(y,d)$ to estimate $\widehat{p}_d(y|x)$.

Formalizing this intuition, we introduce the problem of unsupervised learning under *Latent Label Shift* (LLS). Here, we assume access to a collection of domains $d \in \{1, \ldots, r\}$, where the mixture proportions $p_d(y)$ vary across domains but the class conditional distribution $p(x|y)$ is domain-invariant. Our goals are to recover the underlying classes up to permutation, and thus to identify both the per-domain mixture proportions $p_d(y)$ and optimally adapted per-domain classifiers $p_d(y|x)$. The essential feature of our setup is that only the true $y$'s, as characterized by their class-conditional distributions $p(x|y)$, could account for the observed shifts in $p_d(x)$. We prove that under mild assumptions, knowledge of this underlying structure is sufficient for inducing the full set of categories.

First, we focus on the *tabular setting*, demonstrating that when the input space is discrete and finite, LLS is isomorphic to topic modeling [9]. Here, each distinct input $x$ maps to a *word* each *latent label* $y$ maps to a *topic* and each domain $d$ maps to a *document*. In this case, we can apply standard identification results for topic modeling [22, 5, 30, 38, 14] that rely only on the existence of anchor words within each topic (i.e., for each label $y$ there is at least one $x$ in the support of $y$, that is not in the support of any $y' \neq y$). Here, standard methods based on Non-negative Matrix Factorization (NMF) can recover each domain's underlying mixture proportion $p_d(y)$ and optimal predictor $p_d(y|x)$ [22, 38, 30]. However, the restriction to discrete inputs, while appropriate for topic modeling, proves restrictive when our interests extend to high-dimensional continuous input spaces.

Then, to handle high-dimensional inputs, we propose *Discriminate-Discretize-Factorize-Adjust (DDFA)*, a general framework that proceeds in the following steps: (i) pool data from all domains to produce a mixture distribution $q(x,d)$; (ii) train a domain discriminative model $f$ to predict $q(d|x)$; (iii) push all data through $f$, cluster examples in the pushforward distribution, and tabularize the data based on cluster membership; (iv) solve the resulting discrete topic modeling problem (e.g., via NMF), estimating $q(y,d)$ up to permutation of the latent labels; (v) combine the predicted $q(d|x)$ and $q(y,d)$ to estimate $p_d(y)$ and $p_d(y|x)$. In developing this approach, we draw inspiration from recent works on distribution shift and learning from positive and unlabeled data that (i) leverage black box predictors to perform dimensionality reduction [46, 25, 26]; and (ii) work with *anchor sets*, separable subsets of continuous input spaces that belong to only one class's support [65, 49, 23, 7, 26].

Our key theoretical result shows that domain discrimination ($q(d|x)$) provides a sufficient representation for identifying all parameters of interest. Given oracle access to $q(d|x)$ (which is identified without labels), our procedure is consistent. Our analysis reveals that the true $q(d|x)$ maps all points in the same anchor set to a single point mass in the push-forward distribution. This motivates our practical approach of discretizing data by hunting for tight clusters in $q(d|x)$ space.

In semi-synthetic experiments, we adapt existing image classification benchmarks to the LLS setting, sampling without replacement to construct collections of label-shifted domains. We note that training a domain discriminator classifier is a difficult task, and find that warm starting the initial layers of our model with pretrained weights from unsupervised approaches can significantly boost performance. We show that warm-started DDFA outperforms competitive unsupervised approaches on CIFAR-10 and CIFAR-20 when domain marginals $p_d(y)$ are sufficiently sparse. In particular, we observe improvements of as much as $30\%$ accuracy over a recent high-performing unsupervised method on CIFAR-20. Further, on subsets of FieldGuide dataset, where similarity between species and diversity within a species leads to failure of unsupervised learning, we show that DDFA recovers the true

distinctions. To be clear, these are not apples-to-apples comparisons: our methods are specifically tailored to the LLS setting. The takeaway is that the structure of the *LLS* setting can be exploited to outperform the best unsupervised learning heuristics.

## 2 Related Work

**Unsupervised Learning**  Standard unsupervised learning approaches for discovering labels often rely on similarity in the original data space [50, 62]. While distances in feature space become meaningless for high-dimensional data, deep learning researchers have turned to similarity in a representation space learned via self-supervised contrastive tasks [53, 21, 29, 12], or similarity in a feature space learned end-to-end for a clustering task [10, 11, 57, 72]. Our problem setup closely resembles independent component analysis (ICA), where one seeks to identify statistically independent signal components from mixtures [39]. However, ICA's assumption of statistical independence among the components does not generally hold in our setup. In topic modeling [9, 5, 38, 14, 56], documents are modeled as mixtures of topics, and topics as categorical distributions over a finite vocabulary. Early topic models include the well-known Latent Dirichlet Allocation (LDA) [9], which assumes that topic mixing coefficients are drawn from a Dirichlet distribution, along with papers with more relaxed assumptions on the distribution of topic mixing coefficients (e.g., pLSI) [37, 56]. The topic modeling literature often draws on Non-negative Matrix Factorization (NMF) methods [54, 66], which decompose a given matrix into a product of two matrices with non-negative elements [20, 18, 28, 31]. In both Topic Modeling and NMF, a fundamental problem has been to characterize the precise conditions under which the system is uniquely identifiable [22, 5, 38, 14]. The anchor condition (also referred to as separability) is known to be instrumental for identifying topic models [5, 14, 38, 22]. In this work, we extend these ideas, leveraging separable subsets of each label's support (the anchor sets) to produce anchor words in the discretized problem. Existing methods have attempted to extend latent variable modeling to continuous input domains by making assumptions about the functional forms of the class-conditional densities, e.g., restricting to Gaussian mixtures [62, 60]. Recent work in non-parametric mixture modeling [63, 73] employs the grouping of unlabeled samples to better identify unknown mixtures, but unlike our setting, each group is known to be sampled only from a single mixture component. A second line of approach involves finding an appropriate discretization of the continuous space [71].

**Distribution Shift under the Label Shift Assumption**  The label shift assumption, where $p_d(y)$ can vary but $p(x|y)$ cannot, has been extensively studied in the domain adaptation literature [64, 68, 77, 46, 33, 25] and also features in the problem of learning from positive and unlabeled data [24, 8, 26]. For both problems, many classical approaches suffer from the curse of dimensionality, failing in the settings where deep learning prevails. Our solution strategy draws inspiration from recent work on label shift [46, 1, 6, 25] and PU learning [8, 49, 65, 26, 27] that leverage black-box predictors to produce sufficient low-dimensional representations for identifying target distributions of interest (other works leverage black box predictors heuristically [41]). **Key differences:** While PU learning requires identifying *one* new class for which we lack labeled examples provided that the positive class contains an anchor set [26], LLS can identify an arbitrary number of classes (up to permutation) from completely unlabeled data, provided a sufficient number of domains.

**Domain Generalization**  The related problem of Domain Generalization (DG) also addresses learning with data drawn from multiple distributions and where the domain identifiers play a key role [51, 3]. However in DG, we are given *labeled* data from multiple domains, and our goal is to learn a classifier that can generalize to new domains. By contrast, in LLS, we work with unlabeled data only, leveraging the problem structure to identify the underlying labels.

**Learning with Label Noise (LLN)**  Another problem type that can be viewed under the LLS lens is Learning with Label Noise (LLN) [58, 32, 45, 55, 61, 70, 47, 48]. This connection can be established as follows: Consider the observed noisy labels to act as domains (see 1). We then have (i) different distributions over the true labels conditioned on the noisy label; (ii) When conditioned on the true label, the distribution over the input space is independent of the noisy label. These observations indicate the satisfaction of LLS setup (see 1) in LLN, a key difference being the LLN literature typically assumes the distribution over the noisy label has the highest mass at $\widehat{y} = i$ when the true label $y = i$. The connection between mixture proportion estimation and LLN has been identified in [65]. Many methods have been proposed to tackle the LLN problem: (i) loss correction and surrogate risk minimization [58, 32, 65, 52]; (ii) leveraging the predictions of Deep Networks to assign soft labels, as

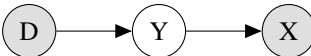

Figure 2: Relationship under $Q$ between observed $D$, observed $X$, and latent $Y$.

teacher-student models or in MixUp like augmentations [34, 13, 61, 70, 2]; (iii) tapping into the early learning of meaningful patterns by Deep Networks [45, 47, 67]. Of particular relevance is [58], which estimates the Noise Transition Matrix (NTM) and then uses this estimate to retrain under a corrected loss function. [58] shows that "perfect examples" allow for the identification of the NTM (similar to the anchor subdomain assumption which appears in this work). However, [58] use the assumption that when the true label takes a certain value, the highest probability mass for the noisy label is also at the same value—an assumption not always satisfied in our setting. Further, in the LLN problem, the number of domains and classes are always equal, rendering LLN methods such as [58] incapable of using the information provided by additional domains present in non-LLN settings. Our method can leverage domain information to provide a domain calibrated classifier, but [58] does not exploit the possible availability of a noisy label at inference. In recent work, [76] use methods from [58] to connect LLN to learning from multiple bags of samples for which the label proportions are known (similar to our setting, but with more information available in the form of these label proportions).

## 3 Latent Label Shift Setup

**Notation** For a vector $v \in \mathbb{R}^p$, we use $v_j$ to denote its $j^{\text{th}}$ entry, and for an event $E$, we let $\mathbb{I}[E]$ denote the binary indicator of the event. By $|A|$, we denote the cardinality of set $A$. With $[n]$, we denote the set $\{1, 2, \ldots, n\}$. We use $[A]_{i,j}$ to access the element at $(i, j)$ in $A$. Let $\mathcal{X}$ be the input space and $\mathcal{Y} = \{1, 2, \ldots, k\}$ be the output space for multiclass classification. We assume throughout this work that the number of true classes $k$ is known. Throughout this paper, we use capital letters to denote random variables and small case letters to denote the corresponding values they take. For example, by $X$ we denote the input random variable and by $x$, we denote a value that $X$ may take.

We now formally introduce the problem of unsupervised learning under LLS. In LLS, we assume that we observe unlabeled data from $r$ domains. Let $\mathcal{R} = \{1, 2, \ldots, r\}$ be the set of domains. By $p_d$, we denote the probability density (or mass) function for each domain $d \in \mathcal{R}$.

**Definition 1** (Latent label shift). *We observe data from $r$ domains. While the label distribution can differ across the domains, for all $d, d' \in \mathcal{R}$ and for all $(x, y) \in \mathcal{X} \times \mathcal{Y}$, we have $p_d(x|y) = p_{d'}(x|y)$.*

Simply put, Definition 1 states that the conditional distribution $p_d(x|y)$ remains invariant across domains, i.e., they satisfy the label shift assumption. Thus, we can drop the subscript on this factor, denoting all $p_d(x|y)$ by $p(x|y)$. Crucially, under LLS, $p_d(y)$ can vary across different domains. Under LLS, we observe unlabeled data with domain label $\{(x_1, d_1), (x_2, d_2), \ldots, (x_n, d_n)\}$. Our goal breaks down into two tasks. Up to permutation of labels, we aim to (i) estimate the label marginal in each domain $p_d(y)$; and (ii) estimate the optimal per-domain predictor $p_d(y|x)$.

**Mixing distribution Q** A key step in our algorithm will be to train a domain discriminative model. Towards this end we define $Q$, a distribution over $\mathcal{X} \times \mathcal{Y} \times \mathcal{R}$, constructed by taking a uniform mixture over all domains. By $q$, we denote the probability density (or mass) function of $Q$. Define $Q$ such that $q(x, y|D = d) = p_d(x, y)$, i.e., when we condition on $D = d$ we recover the joint distribution over $\mathcal{X} \times \mathcal{Y}$ specific to that domain $d$. For all $d \in \mathcal{R}$, we define $\gamma_d = q(d)$, i.e., the prevalence of each domain in our distribution $Q$. Notice that $q(x, y)$ is a mixture over the distributions $\{p_d(x, y)\}_{d \in \mathcal{R}}$, with $\{\gamma_d\}_{d \in \mathcal{R}}$ as the corresponding mixture coefficients. Under LLS (Definition 1), $X$ does not depend on $D$ when conditioned on $Y$ (Fig. 2).

**Additional notation for the discrete case** To begin, we setup notation for discrete input spaces with $|\mathcal{X}| = m$. Without loss of generality, we assume that $\mathcal{X} = \{1, 2, \ldots, m\}$. The label shift assumption allows us to formulate the label marginal estimation problem in matrix form. Let $\mathbf{Q}_{X|D}$ be an $m \times r$ matrix such that $[\mathbf{Q}_{X|D}]_{i,d} = p_d(X = i)$, i.e., the $d$-th column of $\mathbf{Q}_{X|D}$ is $p_d(x)$. Let $\mathbf{Q}_{X|Y}$ be an $m \times k$ matrix such that $[\mathbf{Q}_{X|Y}]_{i,j} = p(X = i|Y = j)$, the $j$-th column is a distribution over $X$ given $Y = j$. Similarly, define $\mathbf{Q}_{Y|D}$ as a $k \times r$ matrix whose $d$-th column is the domain marginal $p_d(y)$. Now with Definition 1, we have $p_d(x) = \sum_y p_d(x, y) = \sum_y p_d(x|y)p_d(y) = \sum_y p(x|y)p_d(y)$. Since this is true $\forall d \in \mathcal{R}$, we can express this in a matrix form as $\mathbf{Q}_{X|D} = \mathbf{Q}_{X|Y}\mathbf{Q}_{Y|D}$.

**Additional assumptions**   Before we present identifiability results for the LLS problem, we introduce four additional assumptions required throughout the paper:

A.1  There are at least as many domains as classes, i.e., $|\mathcal{R}| \geqslant |\mathcal{Y}|$.

A.2  The matrix formed by label marginals (as columns) across different domains is full-rank, i.e., $\text{rank}(\mathbf{Q}_{Y|D}) = k$.

A.3  Equal representation of domains, i.e., for all $d \in \mathcal{R}, \gamma_d = 1/r$.

A.4  Fix $\epsilon > 0$. For all $y \in \mathcal{Y}$, there exists a unique subdomain $A_y \subseteq \mathcal{X}$, such that $q(A_y) \geqslant \epsilon$ and $x \in A_y$ if and only if the following conditions are satisfied: $q(x|y) > 0$ and $q(x|y') = 0$ for all $y' \in \mathcal{Y}\backslash\{y\}$. We refer to this assumption as the $\epsilon$-anchor sub-domain condition.

We now comment on the assumptions. A.1–A.2 are benign, these assumptions just imply that the matrix $\mathbf{Q}_{Y|D}$ is full row rank. Without loss of generality, A.3 can be assumed when dealing with data from a collection of domains. When this condition is not satisfied, one could just re-sample data points uniformly at random from each domain $d$. Intuitively, A.4 states that for each label $y \in \mathcal{Y}$, we have some subset of inputs that only belong to that class $y$. To avoid vanishing probability of this subset, we ensure at least $\epsilon$ probability mass in our mixing distribution $Q$. The anchor word condition is related to the positive sub-domain in PU learning, which requires that there exists a subset of $\mathcal{X}$ in which all examples only belong to the positive class [65, 49, 23, 7].

## 4   Theoretical Analysis

In this section, we establish identifiability of LLS problem. We begin by considering the case where the input space is discrete and formalize the isomorphism to topic modeling. Then we establish the identifiability of the system in this discrete setting by appealing to existing results in topic modeling [38]. Finally, extending results from discrete case, we provide novel analysis to establish our identifiability result for the continuous setting.

**Isomorphism to topic modeling**   Recall that for the discrete input setting, we have the matrix formulation: $\mathbf{Q}_{X|D} = \mathbf{Q}_{X|Y}\mathbf{Q}_{Y|D}$. Consider a corpus of $r$ documents, consisting of terms from a vocabulary of size $m$. Let $\mathbf{D}$ be an $\mathbb{R}^{m \times r}$ matrix representing the underlying corpus. Each column of $\mathbf{D}$ represents a document, and each row represents a term in the vocabulary. Each element $[\mathbf{D}]_{i,j}$ represents the frequency of term $i$ in document $j$. Topic modeling [9, 37, 38, 5] considers each document to be composed as a mixture of $k$ topics. Each topic prescribes a frequency with which the terms in the vocabulary occur given that topic. Further, the proportion of each topic varies across documents with the frequency of terms given topic remaining invariant.

We can state the topic modeling problem as: $\mathbf{D} = \mathbf{CW}$, where $\mathbf{C}$ is an $\mathbb{R}^{m \times k}$ matrix, $[\mathbf{C}]_{i,j}$ represents the frequency of term $i$ given topic $j$, and $\mathbf{W}$ is an $\mathbb{R}^{k \times r}$ matrix, where $[\mathbf{W}]_{i,j}$ represents the proportion of topic $i$ in document $j$. Note that all three matrices are column normalized. The isomorphism is then between document and domain, topic and label, term and input sample, i.e., $\mathbf{D} = \mathbf{CW} \equiv \mathbf{Q}_{X|D} = \mathbf{Q}_{X|Y}\mathbf{Q}_{Y|D}$. In both the cases, we are interested in decomposing a known matrix into two unknown matrices. This formulation is examined as a non-negative matrix factorization problem with an added simplicial constraint on the columns (columns sum to 1) [4, 30].

Identifiability of the topic modeling problem is well-established [22, 5, 30, 38, 14]. We leverage the isomorphism to topic modeling to extend this identifiability condition to our LLS setting. We formalize the adaptation here:

**Theorem 1.** *(adapted from Proposition 1 in Huang et al. [38]) Assume A.1, A.2 and A.4 hold (A.4 in the discrete setting is referred to as the anchor word condition). Then the solution to* $\mathbf{Q}_{X|D} = \mathbf{Q}_{X|Y}\mathbf{Q}_{Y|D}$ *is uniquely identified up to permutation of class labels.*

We refer readers to Huang et al. [38] for a proof of this theorem. Intuitively, Theorem 1 states that if each label $y$ has at least one token in the input space that has support only in $y$, and A.1, A.2 hold, then the solution to $\mathbf{Q}_{X|Y}, \mathbf{Q}_{Y|D}$ is unique. Furthermore, under this condition, there exist algorithms that can recover $\mathbf{Q}_{X|Y}, \mathbf{Q}_{Y|D}$ within some permutation of class labels [38, 30, 4, 5].

**Extensions to the continuous case**   We will prove identifiability in the continuous setting, when $\mathcal{X} = \mathbb{R}^p$ for some $p \geqslant 1$. In addition to A.1–A.4, we make an additional assumption that we have

oracle access to $q(d|x)$, i.e., the true domain discriminator for mixture distribution $Q$. This is implied by assuming access to the marginal $q(x, d)$ from which we observe our samples. Formally, we define a push forward function $f$ such that $[f(x)]_d = q(d|x)$, then push the data forward through $f$ to obtain outputs in $\Delta^{r-1}$. In the proof of Theorem 2, we will show that these outputs can be discretized in a fashion that maps anchor subdomains to anchor words in a tabular, discrete setting. We separately remark that the anchor word outputs are in fact extreme corners of the convex polytope in $\Delta^{r-1}$ which encloses all $f(x)$ mass; we discuss this geometry further in App. F. After constructing the anchor word discretization, we appeal to Theorem 1 to recover $\mathbf{Q}_{Y|D}$. Given $\mathbf{Q}_{Y|D}$, we show that we can use Bayes' rule and the LLS condition (Definition 1) to identify the distribution $q(y|x, d) = p_d(y|x)$ over latent variable $y$. We formalize this in the following theorem:

**Theorem 2.** *Let the distribution $Q$ over random variables $X, Y, D$ satisfy Assumptions A.1–A.4. Assuming access to the joint distribution $q(x, d)$, and knowledge of the number of true classes $k$, we show that the following quantities are identifiable: (i) $\mathbf{Q}_{Y|D}$, (ii) $q(y|X = x)$, for all $x \in \mathcal{X}$ that lies in the support (i.e. $q(x) > 0$); and (iii) $q(y|X = x, D = d)$, for all $x \in \mathcal{X}$ and $d \in \mathcal{R}$ such that $q(x, d) > 0$.*

Before presenting a proof sketch for Theorem 2, we first present key lemmas (we include their proofs in App. A).

**Lemma 1.** *Under the same assumptions as Theorem 2, the matrix $\mathbf{Q}_{Y|D}$ and $f(x) = q(d|x)$ uniquely determine $q(y|x)$ for all $y \in \mathcal{Y}$ and $x \in \mathcal{X}$ such that $q(x) > 0$.*

Lemma 1 states that given matrix $\mathbf{Q}_{Y|D}$ and oracle domain discriminator, we can uniquely identify $q(y|x)$. In particular, we show that for any $x \in \mathcal{X}$, $q(d|x)$ can be expressed as a convex combination of the $k$ columns of $\mathbf{Q}_{D|Y}$ (which is computed from $\mathbf{Q}_{Y|D}$ and is column rank $k$) and the coefficients of the combination are $q(y|x)$. Combining this with the linear independence of the columns of $\mathbf{Q}_{D|Y}$, we show that these coefficients are unique. In the following lemma, we show how the identified $q(y|x)$ can then be used to identify $q(y|x, d)$:

**Lemma 2.** *Under the same assumptions as Theorem 2, for all $y \in \mathcal{Y}$, and $x \in \mathcal{X}$ such that $q(x, d) > 0$. the matrix $\mathbf{Q}_{Y|D}$ and $q(y|x)$ uniquely determine $q(y|x, d)$.*

To prove Lemma 2, we show that we can combine the conditional distribution over the labels given a sample $x \in \mathcal{X}$ with the prior distribution of the labels in each domain to determine the posterior distribution over labels given the sample $x$ and the domain of interest. Next, we introduce a key property of the domain discriminator classifier $f$:

**Lemma 3.** *Under the same assumptions as Theorem 2, for all $x, x'$ in anchor sub-domain, i.e., $x, x' \in A_y$ for a given label $y \in \mathcal{Y}$, we have $f(x) = f(x')$. Further, for any $y \in \mathcal{Y}$, if $x \in A_y, x' \notin A_y$, then $f(x) \neq f(x')$.*

Lemma 3 implies that the oracle domain discriminator $f$ maps all points in an anchor subdomain, and only those points in that anchor subdomain to the same point in $f(x) = q(d|x)$ space. We can now present a proof sketch for Theorem 2 (full proof in App. B):

*Proof sketch of Theorem 2.* The key idea of the proof lies in proposing a discretization such that some subset of anchor subdomains for each label $y$ in the continuous space map to distinct anchor words in discrete space. In particular, if there exists a discretization of the continuous space $\mathcal{X}$ that for any $y \in \mathcal{Y}$, maps all $x \in A_y$ to the same point in the discrete space, but no $x \notin A_y$ maps to this point, then this point serves as an anchor word. From Lemma 3, we know that all the $x \in A_y$ and only the $x \in A_y$ get mapped to specific points in the $f(x)$ space. Pushing all the $x \in \mathcal{X}$ through $f$, we know from A.4 that there exists $k$ point masses of size $\epsilon$, one for each $f(A_y)$ in the $f(x) = q(d|x)$ space. We can now inspect this space for point masses of size at least $\epsilon$ to find at most $\mathcal{O}(1/\epsilon)$ such point masses among which are contained the $k$ point masses corresponding to the anchor subdomains. Discretizing this space by assigning each point mass to a group (and non-point masses to a single additional group), we have $k$ groups that have support only in one $y$ each. Thus, we have achieved a discretization with anchor words. Further, since the discrete space arises from a pushforward of the continuous space through $f$, the discrete space also satisfies the latent label shift assumption A.1. We now use Theorem 1 to claim identifiability of $\mathbf{Q}_{Y|D}$. We then use Lemmas 1 and 2 to prove parts (ii) and (iii).

## 5 DDFA Framework

Motivated by our identifiability analysis, in this section, we present an algorithm to estimate $\mathbf{Q}_{Y|D}, q(y|x)$, and $q(y|x,d)$ when $X$ is continuous by exploiting domain structure and approximating the true domain discriminator $f$. Intuitively, $q(y|x,d)$ is the domain specific classifier $p_d(y|x)$ and $q(y|x)$ is the classifier for data from aggregated domains. $\mathbf{Q}_{Y|D}$ captures label marginal for individual domains. A naive approach would be to aggregate data from different domains and exploit recent advancements in unsupervised learning [72, 57, 10, 11]. However, aggregating data from multiple domains loses the domain structure that we hope to leverage. We highlight this failure mode of the traditional unsupervised clustering method in Sec. 6. We remark that DDFA draws heavy inspiration from the proof of Theorem 2, but we do not present a guarantee that the DDFA solution will converge to the identifiable solution. This is primarily due to the K-means clustering heuristic we rely on, which empirically offers effective noise tolerance, but theoretically has no guarantee of correct convergence.

**Discriminate**  We begin Algorithm 1 by creating a split of the unlabeled samples into the training and validation sets. Using the unlabeled data samples and the domain that each sample originated from, we first train a domain discriminator $\hat{f}$. The domain discriminator outputs a distribution over domains for a given input. This classifier is trained with cross-entropy loss to predict the domain label of each sample on the training set. With unlimited data, the minimizer of this loss is the true $f$, as we prove in App. C. To avoid overfitting, we stop training $\hat{f}$ when the cross-entropy loss on the validation set stops decreasing. Note that here the validation set also only contains domain indices (and no information about true class labels).

**Discretize**  We now push forward all the samples from the training and validation sets through the domain discriminator to get vector $\hat{f}(x_i)$ for each sample $x_i$. In the proof of Theorem 2, we argue that when working with true $f$, and the entire marginal $q(x,d)$, we can choose a discretization satisfying the anchor word assumption by identifying point masses in the distribution of $f(x)$ and filtering to include those of at least $\epsilon$ mass. In the practical setting, because we have only a finite set of data points and a noisy $\hat{f}$, we use clustering to approximately find point masses. We choose $m \geqslant k$ and recover $m$ clusters with any standard clustering procedure (e.g. K-means). This clustering procedure is effectively a useful, but imperfect heuristic: if the noise in $\hat{f}$ is sufficiently small and the clustering sufficiently granular, we hope that our $m$ discovered clusters will include $k$ *pure* clusters, each of which only contains data points from a different anchor subdomain which are tightly packed around the true $f(A_y)$ for the corresponding label $y$. Clustering in this space is superior to a naive clustering on the input space because close proximity in this space indicates similarity in $q(d|x)$.

Let us denote the learned clustering function as $c$, where $c(x)$ is the cluster assigned to a datapoint $x$. We now leverage the cluster id $c(x_i)$ of each sample $x_i$ to discretize samples into a finite discrete space $[m]$. Combining the cluster id with the domain source $d_i$ for each sample, we estimate $\hat{\mathbf{Q}}_{c(X)|D}$ by simply computing, for each domain, the fraction of its samples assigned to each cluster.

**Factorize**  We apply an NMF algorithm to $\hat{\mathbf{Q}}_{c(X)|D}$ to obtain estimates of $\hat{\mathbf{Q}}_{c(X)|Y}$ and $\hat{\mathbf{Q}}_{Y|D}$.

**Adjust**  We begin Algorithm 2 by considering a test point $(x', d')$. To make a prediction, if we had access to oracle $f$ and true $\mathbf{Q}_{Y|D}$, we could precisely compute $q(y|x')$ (Lemma 1). However, in place of these true quantities, we plug in the estimates $\hat{f}$ and $\hat{\mathbf{Q}}_{Y|D}$. Since our estimates contain noise, the estimate $\hat{q}(y|x')$ is found by left-multiplying $\hat{f}(x')$ with the pseudo-inverse of $\hat{\mathbf{Q}}_{D|Y}$, as opposed to solving a sufficient system of equations. As our estimates $\hat{f}$ and $\hat{\mathbf{Q}}_{D|Y}$ approach the true values, the projection of $\hat{f}(x')$ into the column space of $\hat{\mathbf{Q}}_{D|Y}$ tends to $\hat{f}(x')$ itself, so the pseudo-inverse approaches the true solution. Now we can use the constructive procedure introduced in the proof of Lemma 2 to compute the plug-in estimate $\hat{q}(y|x', d') = \hat{p}_{d'}(y|x')$.

## 6 Experiments

Experiment code is available at `https://github.com/acmi-lab/Latent-Label-Shift-DDFA`.

**Baselines**  We select the unsupervised classification method SCAN, as a high-performing competitive baseline [72]. SCAN pretrains a ResNet [35] backbone using SimCLR [12] and MoCo [36] setups (pretext tasks). SCAN then trains a clustering head to minimize the SCAN loss (refer [72] for more

---

**Algorithm 1** DDFA Training

---

**input** $k \geqslant 1, r \geqslant k, \{(x_i, d_i)\}_{i \in [n]} \sim q(x, d)$, A class of functions $\mathcal{F}$ from $\mathbb{R}^p \to \mathbb{R}^r$

  1: Split into train set $T$ and validation set $V$

  2: Train $\hat{f} \in \mathcal{F}$ to minimize cross entropy loss for predicting $d|x$ on $T$ with early stopping on $V$

  3: Push all $\{x_i\}_{i \in [n]}$ through $\hat{f}$

  4: Train clustering algorithm on the n points $\{\hat{f}(x_i)\}_{i \in [n]}$, obtain $m$ clusters.

  5: $c(x_i) \leftarrow$ Cluster id of $\hat{f}(x_i)$

  6: $\hat{q}(c(X) = a|D = b) \leftarrow \frac{\sum_{i \in [n]} \mathbb{I}[c(x_i)=a,\ d_i=b]}{\sum_{j \in [n]} \mathbb{I}[d_j=b]}$

  7: Populate $\hat{\mathbf{Q}}_{c(X)|D}$ as $[\hat{\mathbf{Q}}_{c(X)|D}]_{a,b} \leftarrow \hat{q}(c(X) = a|D = b)$

  8: $\hat{\mathbf{Q}}_{c(X)|Y}, \hat{\mathbf{Q}}_{Y|D} \leftarrow$ NMF $(\hat{\mathbf{Q}}_{c(X)|D})$

**output** $\hat{\mathbf{Q}}_{Y|D}, \hat{f}$

---

---

**Algorithm 2** DDFA Prediction

---

**input** $\hat{\mathbf{Q}}_{Y|D}, \hat{f}, (x', d') \sim q(x, d)$

  1: Populate $\hat{\mathbf{Q}}_{D|Y}$ as $[\hat{\mathbf{Q}}_{D|Y}]_{d,y} \leftarrow \frac{[\hat{\mathbf{Q}}_{Y|D}]_{y,d}}{\sum_{d''=1}^{d''=r} [\hat{\mathbf{Q}}_{Y|D}]_{y,d''}}$

  2: Assign $\hat{q}(y|X = x') \leftarrow \left[ \left( \hat{\mathbf{Q}}_{D|Y} \right)^{\dagger} \hat{f}(x') \right]_y$

  3: Assign $\hat{q}(y|X = x', D = d') \leftarrow \dfrac{[\hat{\mathbf{Q}}_{D|Y}]_{d',y} \hat{q}(y|X = x')}{\sum\limits_{y'' \in [k]} [\hat{\mathbf{Q}}_{D|Y}]_{d',y''} \hat{q}(y''|X = x')}$

  4: $y_{\text{pred}} \leftarrow \arg\max_{y \in [k]} \hat{q}(y|X = x', D = d')$

**output** : $\hat{q}(y|X = x', D = d') = \hat{p}_{d'}(y|x'), \hat{q}(y|X = x'), y_{\text{pred}}$

---

details) [3]. We do not use the SCAN self-labeling step. We make sure to evaluate SCAN on the same potentially class-imbalanced test subset we create for each experiment. Since SCAN is fit on a superset of the data DDFA sees, we believe this gives a slight data advantage to the SCAN baseline (although we acknowledge that the class balance for SCAN training is also potentially different from its evaluation class balance). To evaluate SCAN, we use the public pretrained weights available for CIFAR-10, CIFAR-20, and ImageNet-50. We also train SCAN ourselves on the train and validation portions of the FieldGuide2 and FieldGuide28 datasets with a ResNet18 backbone and SimCLR pretext task. We replicate the hyperparameters used for CIFAR training.

**Datasets** First we examine standard multiclass image datasets CIFAR-10, CIFAR-20 [43], and ImageNet-50 [19] containing images from 10, 20, and 50 classes respectively. Images in these datasets typically focus on a single large object which dominates the center of the frame, so unsupervised classification methods which respond strongly to similarity in visual space are well-suited to recover true classes up to permutation. These datasets are often believed to be separable (i.e., single true label applies to each image), so every example falls in an anchor subdomain (satisfying A.4).

Motivated by the application of LLS problem, we consider the FieldGuide dataset [4], which contains images of moths and butterflies. The true classes in this dataset are species, but each class contains images taken in immature (caterpillar) and adult stages of life. Based on the intuition that butterflies from a given species look more like butterflies from other species than caterpillars from their own species, we hypothesize that unsupervised classification will learn incorrect class boundaries which distinguish caterpillars from butterflies, as opposed to recovering the true class boundaries. Due to high visual similarity between members of different classes, this dataset may indeed have slight overlap between classes. However, we hypothesize that anchor subdomain still holds, i.e., there exist some images from each class that could only come from that class. Additionally, if we have access to data from multiple domains, it is natural to assume that within each domain the relative distribution of caterpillar to adult stages of each species stay relatively constant as compared to

---

[3]SCAN code: `https://github.com/wvangansbeke/Unsupervised-Classification`

[4]FieldGuide: `https://sites.google.com/view/fgvc6/competitions/butterflies-moths-2019`

Table 1: *Results on CIFAR-20.* Each entry is produced with the averaged result of 3 different random seeds. With DDFA (RI) we refer to DDFA with randomly initialized backbone. With DDFA (SI) we refer to DDFA's backbone initialized with SCAN. Note that in DDFA (SI), we do not leverage SCAN for clustering. $\alpha$ is the Dirichlet parameter used for generating label marginals in each domain, $\kappa$ is the maximum allowed condition number of the generated $\mathbf{Q}_{Y|D}$ matrix, $r$ is number of domains.

| r | Approaches | $\alpha : 0.5, \ \kappa : 8$ | | $\alpha : 3, \ \kappa : 12$ | | $\alpha : 10, \ \kappa : 20$ | |
|---|---|---|---|---|---|---|---|
| | | Test acc | $\mathbf{Q}_{Y|D}$ err | Test acc | $\mathbf{Q}_{Y|D}$ err | Test acc | $\mathbf{Q}_{Y|D}$ err |
| 20 | SCAN | 0.454 | 0.059 | 0.421 | 0.051 | **0.436** | 0.038 |
| | DDFA (RI) | 0.520 | 0.041 | 0.357 | 0.043 | 0.187 | 0.051 |
| | DDFA (SI) | **0.852** | **0.015** | **0.548** | **0.026** | 0.354 | **0.036** |
| 25 | SCAN | 0.458 | 0.059 | 0.455 | 0.048 | 0.440 | 0.037 |
| | DDFA (RI) | 0.525 | 0.042 | 0.310 | 0.051 | 0.182 | 0.053 |
| | DDFA (SI) | **0.819** | **0.021** | **0.707** | **0.022** | **0.502** | **0.030** |
| 30 | SCAN | 0.456 | 0.059 | 0.441 | 0.050 | 0.437 | 0.037 |
| | DDFA (RI) | 0.506 | 0.045 | 0.256 | 0.058 | 0.088 | 0.076 |
| | DDFA (SI) | **0.845** | **0.020** | **0.688** | **0.027** | **0.531** | **0.029** |

prevalence of different species. We create two subsets of this dataset: FieldGuide2, with two species, and FieldGuide28, with 28 species.

**LLS Setup** The full sampling procedure for semisynthetic experiments is described in App. D. Roughly, we sample $p_d(y)$ from a symmetric Dirichlet distribution with concentration $\alpha/k$, where $k$ is the number of classes and $\alpha$ is a generation parameter that adjusts the difficulty of the synthetic problem, and enforce maximum condition number $\kappa$ on $\mathbf{Q}_{\mathbf{Y}|\mathbf{D}}$. Small $\alpha$ and small $\kappa$ encourages sparsity in $\mathbf{Q}_{\mathbf{Y}|\mathbf{D}}$, so each label tends to only appear in a few domains. Larger parameters encourages $p_d(y)$ to tend toward uniform. We draw from test, train, and valid datasets without replacement to match these distributions, but discard some examples due to class imbalance.

**Training and Evaluation** The algorithm uses train and validation data consisting of pairs of images and domain indices. We train ResNet50 [35] (with added dropout) on images $x_i$ with domain indices $d_i$ as the label, choose best iteration by valid loss, pass all training and validation data through $\widehat{f}$, and cluster pushforward predictions $\widehat{f}(x_i)$ into $m \geqslant k$ clusters with Faiss K-Means [42]. We compute the $\widehat{\mathbf{Q}}_{c(X)|D}$ matrix and run NMF to obtain $\widehat{\mathbf{Q}}_{c(X)|Y}$, $\widehat{\mathbf{Q}}_{Y|D}$. To make columns sum to 1, we normalize columns of $\widehat{\mathbf{Q}}_{c(X)|Y}$, multiply each column's normalization coefficient over the corresponding row of $\widehat{\mathbf{Q}}_{Y|D}$ (to preserve correctness of the decomposition), and then normalize columns of $\widehat{\mathbf{Q}}_{Y|D}$. Some NMF algorithms only output solutions satisfying the anchor word property [4, 44, 30]. We found the strict requirement of an exact anchor word solution to lead to low noise tolerance. We therefore use the Sklearn implementation of standard NMF [15, 69, 59].

We instantiate the domain discriminator as ResNet18, and preseed its backbone with SCAN [72] pre-trained weights or [72] contrastive pre-text weights. We denote these models DDFA (SI) and DDFA (SPI) respectively. We predict class labels with Algorithm 2. With the Hungarian algorithm, implemented in [16, 74], we compute the highest true accuracy among any permutation of these labels (denoted "Test acc"). With the same permutation, we reorder rows of $\widehat{P}_{Y|D}$, then compute the average absolute difference between corresponding entries of $\widehat{\mathbf{Q}}_{Y|D}$ and $\mathbf{Q}_{Y|D}$ (denoted "$\mathbf{Q}_{Y|D}$ err").

**Results** On CIFAR-10, we observe that DDFA alone is incapable of matching highly competitive baseline SCAN performance—however, in suitably sparse problem settings (small $\alpha$), it comes substantially close, indicating good recovery of true classes. Due to space constraints, we include CIFAR-10 results in App. E. DDFA (SI) combines SCAN's strong pretrain with domain discrimination fine-tuning to outperform SCAN in the easiest, sparsest setting and *certain* denser settings. On CIFAR-20, baseline SCAN is much less competitive, so our DDFA(SI) dominates baseline SCAN in all settings except the densest (Table 1). These results demonstrate how adding domain information can dramatically boost unsupervised baselines.

On FieldGuide-2, DDFA (SPI) outperforms SCAN baselines in accuracy across all but the densest settings (Table 5); in sparser settings, the accuracy gap is 10-30%. In this dataset, SCAN performs only slightly above chance, reflecting perhaps a total misalignment of learned class distinctions with true species boundaries. We do not believe that SCAN is too weak to effectively detect image groupings on this data; instead we acknowledge that the domain information available to DDFA (SPI) (and not to SCAN) is informative for ensuring recovery of the true class distinction between species as opposed to the visually striking distinction between adult and immature life stages. We do note that, oddly, SCAN tends to better recover the $\mathbf{Q}_{Y|D}$ matrix in harder settings. On FieldGuide-28 (Table 6), DDFA accuracy outperforms SCAN baseline accuracy in all evaluated sparsity levels, with the highest observed accuracy difference ranging above 30-40%.

# 7 Conclusion

Our theoretical results demonstrate that under LLS, we can leverage shifts among previously seen domains to recover labels in a purely unsupervised manner, and our practical instantiation of the DDFA framework demonstrates both (i) the practical efficacy of our approach; and (ii) that generic unsupervised methods can play a key role both in clustering discriminator outputs, and providing weights for initializing the discriminator. We believe our work is just the first step in a new direction for leveraging structural assumptions together with distribution shift to perform unsupervised learning.

## 7.1 Future Work and Limitations

**Assumptions** Our approach is limited by the set of assumptions needed (label shift, as many data domains as true latent classes, known true number of classes $k$, and other assumptions established in A.1-A.4). Future work should aim to relax these assumptions.

**Theory** The work does not include finite sample bounds for the DDFA algorithm. In addition, we do not have a formal guarantee that the clustering heuristic in the Discretize step of DDFA will retrieve pure anchor sub-domain clusters under potentially noisy black-box prediction of $q(d|x)$. This problem is complicated by the difficulty of reasoning about the noise that may be produced by a neural network or other complex non-linear model (acting as the black-box domain discriminator), and by the lack of concrete guarantees that K-means will recover the anchor subdomains among its recovered clusters—especially when classes overlap.

**DDFA Framework** Within the LLS setup, several components of the DDFA framework warrant further investigation: (i) the deep domain discriminator can be enhanced in myriad ways; (ii) for clustering discriminator outputs, we might develop methods specially tailored to our setting to replace the current generic clustering heuristic; (iii) clustering might be replaced altogether with geometrically informed methods that directly identify the corners of the polytope; (iv) the theory of LLS can be extended beyond identification to provide statistical results that might hold when $q(d|x)$ can only be noisily estimated, and when only finite samples are available for the induced topic modeling problem; (v) when the number of true classes $k$ is unknown, we may develop approaches to estimate this $k$.

**Semi-synthetic Experiments** Semi-synthetic experiments present an ideal environment for evaluating under the precise label shift assumption. Evaluating on datasets in which the separation into domains is organic, and the label shift is inherent is an interesting direction for future work.

# Acknowledgements

SG acknowledges Amazon Graduate Fellowship and JP Morgan PhD Fellowship for their support. ZL acknowledges Amazon AI, Salesforce Research, Facebook, UPMC, Abridge, the PwC Center, the Block Center, the Center for Machine Learning and Health, and the CMU Software Engineering Institute (SEI) via Department of Defense contract FA8702-15-D-0002, for their generous support of ACMI Lab's research on machine learning under distribution shift.

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
