## Supplementary Material

## A    Proofs of Lemmas

In this section, we present several new lemmas which are required to prove Theorem 2, and provide proofs. We also provide proofs for Lemmas 1, 2, and 3.

**Lemma 4.** *Let distribution $Q$ over random variables $X, Y, D$ satisfy A.1–A.4. Then for all $y \in \mathcal{Y}$, $q(y) > 0$. That is, all labels have nonzero probability under $Q$.*

*Proof of Lemma 4.* Proof by contradiction. Let $y \in \mathcal{Y}$ with $q(y) = 0$.

$$
\begin{aligned}
q(y) &= \sum_{d \in \mathcal{R}} q(d) q(y|D = d) \\
&= \sum_{d \in \mathcal{R}} \gamma_y q(y|D = d) \\
&= \sum_{d \in \mathcal{R}} \frac{1}{r} q(y|D = d) \\
&= \frac{1}{r} \sum_{d \in \mathcal{R}} q(y|D = d).
\end{aligned}
$$

Since $q(y|D = d) \geqslant 0$ for all $d \in \mathcal{R}$, we see that if $q(y) = 0$, then $q(y|D = d) = 0$ for all $d \in \mathcal{R}$.

Then $[\mathbf{Q}_{Y|D}]_{y,d} = 0$ for all $d \in \mathcal{R}$. Then there is a row (row $y$) in the matrix $\mathbf{Q}_{Y|D}$ in which every entry is 0, so $\mathbf{Q}_{Y|D}$ cannot be full row rank $k$. This violates assumption A.2. Then by contradiction we have shown $q(y) > 0$. □

**Lemma 5.** *Let distribution $Q$ over random variables $X, Y, D$ satisfy Assumptions A.1–A.4. Let $x \in \mathcal{X}$ such that $q(x) > 0$. Then if $x \in A_y$ for some $y \in \mathcal{Y}$, we have that $q(y|X = x) = 1$, and for all $y' \in \mathcal{Y}\backslash\{y\}$, $q(y'|X = x) = 0$. The converse is also true: if $q(y|X = x) = 1$ for some $y \in \mathcal{Y}$ and $q(y'|X = x) = 0 \,\forall y' \in \mathcal{Y}\backslash\{y\}$, then we know that $x \in A_y$.*

*Proof of Lemma 5.* We prove directions one at a time.

**Forward direction.**    Assume $x \in A_y$.

$$
\begin{aligned}
q(x) &= \sum_{y'' \in \mathcal{Y}} q(y'') q(x|Y = y'') \\
&= q(y) q(x|Y = y) + \sum_{y' \in \mathcal{Y}\backslash\{y\}} q(y') q(x|Y = y') \\
&= q(y) q(x|Y = y) + \sum_{y' \in \mathcal{Y}\backslash\{y\}} q(y') \, (0) \\
&= q(y) q(x|Y = y).
\end{aligned}
$$

Recalling $q(x|y) > 0$ (by A.4) and $q(y) > 0$ (by Lemma 4), we know that $q(x) = q(y) q(x|Y = y) > 0$. Then $q(y|X = x) = \dfrac{q(y) q(x|Y = y)}{q(x)} = \dfrac{q(x)}{q(x)} = 1$. Because probabilities sum to 1, $q(y|X = x) + \sum_{y' \in \mathcal{Y}\backslash\{y\}} q(y'|X = x) = 1$. Then because $q(y|X = x) = 1$, we have : $\sum_{y' \in \mathcal{Y}\backslash\{y\}} q(y'|X = x) = 0$. Then for all $y' \in \mathcal{Y}\backslash\{y\}$, it must be that $q(y'|X = x) = 0$. Then we have shown $q(y|X = x) = 1$, and for all $y' \in \mathcal{Y}\backslash\{y\}$, $q(y'|X = x) = 0$.

**Converse.**    Assume $q(y|X = x) = 1$ and for all $y' \in \mathcal{Y}\backslash\{y\}, q(y'|X = x) = 0$. We recall that $q(x) > 0$. Also, $q(y) > 0$ by Lemma 4. Then $q(x|Y = y) = \dfrac{q(y|X = x) q(x)}{q(y)} = \dfrac{(1) q(x)}{q(y)} > 0$. Let $y' \in \mathcal{Y}\backslash\{y\}$. Then $q(x|Y = y') = \dfrac{q(y'|X = x) q(x)}{q(y')} = \dfrac{(0) q(x)}{q(y')} = 0$. Then because $q(x|Y = y) > 0$ and $\forall y' \in \mathcal{Y}\backslash\{y\}$, $q(x|Y = y') = 0$, we see that $x \in A_y$. □

**Lemma 6.** *Let random variables $X, Y, D$ and distribution $Q$ satisfy Assumptions A.1–A.4. Then, the matrix $\mathbf{Q}_{D|Y}$, defined as an $r \times k$ matrix whose elements are $[\mathbf{Q}_{D|Y}]_{i,j} = Q(D = i|Y = j)$, and in which each column is a conditional distribution over the domains given a label, has linearly independent columns. Furthermore, $\mathbf{Q}_{D|Y}$ can be computed directly from only $\mathbf{Q}_{Y|D}$.*

*Proof of Lemma 6.* Let random variables $X, Y, D$ and distribution $Q$ satisfy Assumptions A.1–A.4.

Each $[\mathbf{Q}_{D|Y}]_{d,y} = q(d|Y = y) = \dfrac{q(y|D = d)q(d)}{q(y)} = \dfrac{q(y|D = d)\gamma_d}{q(y)} = \dfrac{q(y|D = d)}{rq(y)}$.

Since each $y$th column of $\mathbf{Q}_{D|Y}$ is a probability distribution that sums to 1, and $rq(y)$ is constant down each $y$th column, we can obtain $\mathbf{Q}_{D|Y}$ by simply taking $\mathbf{Q}_{Y|D}^\top$, in which each $[\mathbf{Q}_{Y|D}^\top]_{d,y} = [\mathbf{Q}_{Y|D}]_{y,d} = q(y|D = d)$, and normalizing the columns so they sum to 1.

The matrix $\mathbf{Q}_{Y|D}$ has linearly independent rows by Assumption A.2. Then $\mathbf{Q}_{Y|D}^\top$ has linearly independent columns. Scaling these columns by a nonzero value does not change their linear independence, so the columns of $\mathbf{Q}_{D|Y}$ are also linearly independent.

Then matrix $\mathbf{Q}_{D|Y}$ has linearly independent columns, and can be computed by taking $\mathbf{Q}_{Y|D}^\top$ and normalizing its columns.

$\square$

**Lemma 7.** *Let random variables $X, Y, D$ and distribution $Q$ satisfy Assumptions A.1–A.4. Let $d \in \mathcal{R}, x \in \mathcal{X}, y \in \mathcal{Y}$. Then $q(d|X = x, Y = y) = q(d|Y = y)$.*

*Proof of Lemma 7.*

$$
\begin{aligned}
q(d|X = x, Y = y) &= \frac{q(x|D = d, Y = y)q(d|Y = y)}{q(x|Y = y)} \\
&= \frac{p_d(x|Y = y)q(d|Y = y)}{q(x|Y = y)} \\
&= \frac{p(x|Y = y)q(d|Y = y)}{q(x|Y = y)} \\
&= \frac{q(x|Y = y)q(d|Y = y)}{q(x|Y = y)} \\
&= q(d|Y = y).
\end{aligned}
$$

$\square$

**Lemma 1.** *We restate this lemma, first presented in Sec. 4, for convenience. Let the distribution $Q$ over random variables $X, Y, D$ satisfy Assumptions A.1–A.4. Then the matrix $\mathbf{Q}_{Y|D}$ and $f(x) = q(d|x)$ uniquely determine $q(y|x)$ for all $y \in \mathcal{Y}$ and $x \in \mathcal{X}$ such that $q(x) > 0$.*

*Proof of Lemma 1.* Let distribution $Q$ over random variables $X, Y, D$ satisfy Assumptions A.1-A.4. Let $x \in \mathcal{X}$ with $q(x) > 0$, and $y \in \mathcal{Y}$. Assume we know $\mathbf{Q}_{Y|D}$ and $[f(x)]_d = q(d|X = x)$. Notice that, for all $x \in \mathcal{X}, d \in \mathcal{R}$,

$$
q(d|X = x) = \sum_{y' \in \mathcal{Y}} q(d|X = x, Y = y')q(y'|X = x).
$$

Now using Lemma 7,

$$
q(d|X = x) = \sum_{y' \in \mathcal{Y}} q(d|Y = y')q(y'|X = x).
$$

Define the vector-valued function $g : \mathcal{X} \to \mathbb{R}^k$ such that $[g(x)]_y = q(y|X = x)$ for all $x \in \text{supp}_Q(X)$. $\mathbf{Q}_{D|Y}$ is a matrix of shape $r \times k$, with $[\mathbf{Q}_{D|Y}]_{i,j} = Q(D = i|Y = j)$. It can be computed from $\mathbf{Q}_{Y|D}$ and has linearly independent columns—both facts shown in Lemma 6.

Then $[f(x)]_d = q(d|X = x) = \mathbf{Q}_{D|Y}[d,:]g(x)$, a product between the $d$th row vector of $\mathbf{Q}_{D|Y}$ and the column vector $g(x)$. Then $f(x) = \mathbf{Q}_{D|Y}g(x)$.

This system is a linear system with $r \geqslant k$ equations. Recalling that $\mathbf{Q}_{D|Y}$ has $k$ linearly independent columns, we can select any $k$ linearly independent rows of $\mathbf{Q}_{D|Y}$ to solve the equation for the true, unique solution for the unknown vector $g(x)$. Another way to describe this is with the pseudo-inverse: $g(x) = (\mathbf{Q}_{D|Y})^\dagger f(x)$. Then we have $[g(x)]_y = q(y|X = x)$ for all $y \in \mathcal{Y}$.

$\square$

**Lemma 2.** *We restate this lemma, first presented in Sec. 4, for convenience. Let the distribution $Q$ over random variables $X, Y, D$ satisfy Assumptions A.1–A.4. Then for all $y \in \mathcal{Y}$, and $x \in \mathcal{X}$ such that $q(x, d) > 0$. the matrix $\mathbf{Q}_{Y|D}$ and $q(y|x)$ uniquely determine $q(y|x, d)$.*

*Proof of Lemma 2.* Let distribution $Q$ over random variables $X, Y, D$ satisfy Assumptions A.1-A.4. Let $x \in \mathcal{X}, d \in \mathcal{R}$ with $q(x, d) > 0$, and $y \in \mathcal{Y}$.

Assume we know matrix $\mathbf{Q}_{Y|D}$ and $q(y'|X = x), \forall y' \in \mathcal{Y}$. We can compute $\mathbf{Q}_{D|Y}$ from $\mathbf{Q}_{Y|D}$ via Lemma 6.

$$
\begin{aligned}
q(y|X = x, D = d) &= \frac{q(y, x, d)}{q(x, d)} \\
&= \frac{q(d|X = x, Y = y)q(y|X = x)q(x)}{q(d|X = x)q(x)}.
\end{aligned}
$$

Using Lemma 7, $q(d|X = x, Y = y) = q(d|Y = y)$. We apply this property.

$$
\begin{aligned}
q(y|X = x, D = d) &= \frac{q(d|Y = y)q(y|X = x)q(x)}{q(d|X = x)q(x)} \\
&= \frac{q(d|Y = y)q(y|X = x)}{q(d|X = x)}.
\end{aligned}
$$

The denominator $q(d|X = x)$ is constant across all values of $y$, so we can write that $q(y|X = x, D = d) \propto q(d|Y = y)q(y|X = x)$ and normalize to find the probability:

$$
q(y|X = x, D = d) = \frac{q(d|Y = y)q(y|X = x)}{\sum\limits_{y' \in \mathcal{Y}} q(d|Y = y')q(y'|X = x)}.
$$

We know $q(d|Y = y)$ as $[\mathbf{Q}_{D|Y}]_{d,y}$, and every $q(d|Y = y')$, where $y' \in \mathcal{Y}\backslash\{y\}$, as $[\mathbf{Q}_{D|Y}]_{d,y'}$. We also know $q(y|X = x)$ and every $q(y'|X = x)$ where $y' \in \mathcal{Y}\backslash\{y\}$, by the precondition assumptions. Then we can compute $q(y|X = x, D = d)$. $\square$

**Lemma 3.** *We restate this lemma, first presented in Sec. 4, for convenience. Let the distribution $Q$ over random variables $X, Y, D$ satisfy Assumptions A.1–A.4. Then for all $x, x'$ in anchor sub-domain, i.e., $x, x' \in A_y$ for a given label $y \in \mathcal{Y}$, we have $f(x) = f(x')$. Further, for any $y \in \mathcal{Y}$, if $x \in A_y, x' \notin A_y$, then $f(x) \neq f(x')$.*

*Proof of Lemma 3.* Let distribution $Q$ over random variables $X, Y, D$ satisfy Assumptions A.1-A.4. Recall $f : \mathbb{R}^p \to \mathbb{R}^r$ is a vector-valued oracle function such that $[f(x)]_d = q(d|X = x)$ for all $x \in \text{supp}_Q(X)$. Also let us recall that $\mathbf{Q}_{D|Y}$ is defined as an $r \times k$ matrix whose elements $[\mathbf{Q}_{D|Y}]_{i,j} = Q(D = i|Y = j)$, and each column is a conditional distribution over the domains given a label. It has linearly independent columns by Lemma 6.

First recognize that for all $d \in \mathcal{R}, x \in \mathcal{X}$ such that $q(x) > 0$,

$$
\begin{aligned}
[f(x)]_d = q(d|X = x) &= \sum_{y'' \in \mathcal{Y}} q(d, y''|X = x). \\
&= \sum_{y'' \in \mathcal{Y}} q(d|Y = y'', X = x)q(y''|X = x).
\end{aligned}
$$

Using Lemma 7, $q(d|X = x, Y = y) = q(d|Y = y)$. We apply this property.

$$[f(x)]_d = q(d|X = x) = \sum_{y'' \in \mathcal{Y}} q(d|Y = y'')q(y''|X = x).$$

Then we can write $f(x) = \sum_{y'' \in \mathcal{Y}} q(y''|X = x)\mathbf{Q}_{D|Y}[:, y'']$, where $\mathbf{Q}_{D|Y}[:, y'']$ is the $y''$th column of $\mathbf{Q}_{D|Y}$. Now we could also rewrite $f(x) = \mathbf{Q}_{D|Y} [Q(Y = 1|X = x) ... Q(Y = k|X = x)]^\top$.

We now prove two key components of the lemma. Let $y \in \mathcal{Y}$. Let $x \in A_y$ such that $q(x) > 0$.

**Points in same anchor sub-domain map together.** Let $x' \in A_y$ such that $q(x') > 0$. We now seek to show that $f(x) = f(x')$. Recall that $x, x' \in A_y$. By Lemma 5, $q(y|X = x) = q(y|X = x') = 1$. Also by lemma 5, $\forall y'' \in \mathcal{Y}\backslash\{y\}, q(y''|X = x) = q(y''|X = x') = 0$. Then for all $y'' \in \mathcal{Y}$, $q(y''|X = x) = q(y''|X = x')$.

Therefore, $\forall d \in \mathcal{R}$,

$$\begin{aligned}
[f(x)]_d = q(d|X = x) &= \sum_{y'' \in \mathcal{Y}} q(d|Y = y'')q(y''|X = x) \\
&= \sum_{y'' \in \mathcal{Y}} q(d|Y = y'')q(y''|X = x') \\
&= q(d|X = x') = [f(x')]_d.
\end{aligned}$$

Then $f(x) = f(x')$.

**Point outside of the anchor sub-domain does not map with points in the anchor sub-domain.** Let $x_0 \notin A_y$ such that $q(x_0) > 0$. We now seek to show that $f(x) \neq f(x_0)$. Because $x_0 \notin A_y$ with $q(x_0) > 0$, and because $A_y$ contains all $x$ such that $q(x) > 0$, $q(y|X = x) = 1$, and $q(y'|X = x) = 0$ for all $y' \in \mathcal{Y}\backslash\{y\}$, then by Lemma 5, it must be that one of the following cases is true:

- **Case 1:** $q(y|X = x_0) \neq 1$
- **Case 2:** $q(y'|X = x_0) > 0$ for some $y' \in \mathcal{Y}\backslash\{y\}$.

In all circumstances, there exists some $y'' \in \mathcal{Y} : q(y''|x_0) \neq q(y''|x)$. Then,

$$[Q(Y = 1|X = x)...Q(Y = k|X = x)]^\top \neq [Q(Y = 1|X = x_0)...Q(Y = k|X = x_0)]^\top.$$

Because $\mathbf{Q}_{D|Y}$ has linearly independent columns (shown in Lemma 6), we now know that

$$\begin{aligned}
f(x) = \mathbf{Q}_{D|Y} [Q(Y = 1|X = x) ... Q(Y = k|X = x)]^\top \\
\neq \mathbf{Q}_{D|Y} [Q(Y = 1|X = x_0) ... Q(Y = k|X = x_0)]^\top = f(x_0).
\end{aligned}$$

So $f(x) \neq f(x_0)$.

$\square$

# B Proof of Theorem 2

**Theorem 2.** *We restate this theorem, first presented in Sec. 4, for convenience. Let the distribution $Q$ over random variables $X, Y, D$ satisfy Assumptions A.1–A.4. Assuming access to the joint distribution $q(x, d)$, and knowledge of the number of true classes $k$, we show that the following quantities are identifiable: (i) $\mathbf{Q}_{Y|D}$, (ii) $q(y|X = x)$, for all $x \in \mathcal{X}$ that lies in the support (i.e. $q(x) > 0$); and (iii) $q(y|X = x, D = d)$, for all $x \in \mathcal{X}$ and $d \in \mathcal{R}$ such that $q(x, d) > 0$.*

*Proof of Theorem 2.* Let distribution $Q$ over random variables $X, Y, D$ satisfy Assumptions A.1-A.4.

Recall $f : \mathcal{X} \to \mathbb{R}^r$ is a vector-valued oracle function such that $[f(x)]_d = q(d|X = x)$ for all $x \in \text{supp}_Q(X)$. It is known because we know the marginal $q(x, d)$. Let $y \in \mathcal{Y}$. Then by Lemma 3, $f$ sends every $x \in A_y$ (and no other $x \notin A_y$) to the same value. We overload notation to denote this as $f(A_y)$. Then $Q(f(X) = f(A_y)) = Q(X \in A_y) \geqslant \epsilon$. Then in the marginal distribution of $f(X)$ with respect to distribution $Q$, there is a distinct point mass on each $f(A_y)$, with mass at least $\epsilon$.

Because we know the marginal $q(x, d)$, we know the marginal $q(x)$, so we can obtain the distribution of $f(X)$ with respect to distribution $Q$. If we analyze the marginal distribution of $f(X)$ with respect to distribution $Q$, and recover all point masses with mass at least $\epsilon$, we can recover no more than $\mathcal{O}(1/\epsilon)$ such points. We set $m \in \mathbb{Z}^+$ so that the number of points we recovered is $m - 1$.

We denote a mapping $\psi : \mathbb{R}^r \to [m]$. This mapping sends each value of $f(x)$ corresponding to a point mass with mass at least $\epsilon$ to a unique index in $\{1, ..., m - 1\}$. It sends any other value in $\mathbb{R}^p$ to $m$. We note that the ordering of the point masses might have $(m - 1)!$ permutations.

Notice that the point mass on each $f(A_y)$ must be recovered among these $m - 1$ masses. Recall that for all $y \in \mathcal{Y}$, $f(x) = f(A_y)$ if and only if $x \in A_y$. Then for all $y \in \mathcal{Y}$, $\psi(f(x)) = \psi(f(A_y))$ if and only if $x \in A_y$, because $\psi$ does not send any other value in $\mathbb{R}^r$ besides $f(A_y)$ to $\psi(f(A_y))$.

For convenience, we now define a mapping $c : \mathcal{X} \to [m]$ such that $c = \psi \circ f$. We will also abuse notation here to denote $c(A_y) = \psi(f(A_y))$. Then $c(X)$ is a discrete, finite random variable that takes values in $[m]$. As $c$ is a pushforward function on $X$, $c(X)$ satisfies the label shift assumption because $X$ does (i.e., when conditioning on $Y$, the distribution of $c(X)$ is domain-invariant).

We might now define a matrix $\mathbf{Q}_{c(X)|D}$ in which each entry $[\mathbf{Q}_{c(X)|D}]_{i,d} = Q(c(X) = i|D = d)$. We recall that we know the number of true classes $k$. Then we know that there is a (possibly unique) unknown decomposition of the following form:

$$q(c(X)|d) = \sum_{y \in \mathcal{Y}} q(c(X)|Y = y, D = d)q(y|D = d)$$

Using the label shift property,

$$q(c(X)|d) = \sum_{y \in \mathcal{Y}} q(c(X)|Y = y)q(y|D = d).$$

To express this decomposition in matrix form, we write $\mathbf{Q}_{c(X)|D} = \mathbf{Q}_{c(X)|Y}\mathbf{Q}_{Y|D}$. Now we make observations about the unknown $\mathbf{Q}_{c(X)|Y}$. For all $y \in \mathcal{Y}$,

$$Q(c(x) = c(A_y)|Y = y) = Q(X \in A_y|Y = y) > 0.$$
$$Q(c(x) = c(A_y)|Y \neq y) = Q(X \in A_y|Y \neq y) = 0.$$

Then for each $y \in \mathcal{Y}$, the row of $\mathbf{Q}_{c(X)|Y}$ with row index $c(A_y)$ is positive in the $y$th column, and zero everywhere else. Restated, for each $y \in \mathcal{Y}$, there is some row with positive entry exactly in $y$th column. This is precisely the anchor word assumption for a discrete, finite random variable. We already know that $\mathbf{Q}_{Y|D}$ is full row-rank, so because $\mathbf{Q}_{c(X)|Y}$ satisfies the anchor word assumption, we can identify $\mathbf{Q}_{Y|D}$ up to permutation of rows by Theorem 1. In other words, when we set the constraint that the recovered $\mathbf{Q}_{c(X)|Y}$ must have $k$ columns and satisfy anchor word and the recovered $\mathbf{Q}_{Y|D}$ must have $k$ rows and be full row-rank, any solution to the decomposition $\mathbf{Q}_{c(X)|D} = \mathbf{Q}_{c(X)|Y}\mathbf{Q}_{Y|D}$ must identify the ground truth $\mathbf{Q}_{Y|D}$, up to permutation of its rows.

$\square$

## C Minimizing Cross-Entropy Loss yields Domain Discriminator

In this section we reason about our choice of cross entropy loss to estimate the domain discriminator. We show that in population, when optimizing over a sufficiently powerful class of functions, the minimizer of the cross entropy loss is the conditional distribution over domains given input $X$.

Define the vector-valued function $z : \mathcal{R} \to \mathbb{R}^r$ such that $z(D)$ is a one-hot random vector of length $r$, such that $[z(D)]_i = 1$, iff $D = i$. Then we write the cross-entropy objective in expectation over the input random variable $X$ and target random variable $D$ as:

$$\mathcal{L} = \mathbb{E}_{(X,D) \sim Q} \left[ - \sum_{i=1}^{i=r} [z(D)]_i \log([f(X)]_i) \right]$$

$$= \mathbb{E}_X \mathbb{E}_{D|X} \left[ - \sum_{i=1}^{i=r} [z(D)]_i \log([f(X)]_i) \right]$$

where the second line follows by splitting the expectations using the law of iterative expectations. For ease of notation we denote conditioning on $X = x$ as just conditioning on $X$. We now move the inner expectation into the summation over the domain random variable and pull the terms that do not depend on $D$ outside this inner expectation:

$$\mathcal{L} = \mathbb{E}_X \left[ - \sum_{i=1}^{i=r} \mathbb{E}_{D|X} \left[ [z(D)]_i \log([f(X)]_i) \right] \right]$$

$$= \mathbb{E}_X \left[ - \sum_{i=1}^{i=r} \log([f(X)]_i) \mathbb{E}_{D|X} \left[ [z(D)]_i \right] \right]$$

Since $[Z(D)]_i$ is a random variable which is 1 when $D = i$ conditioned on $X$, and 0 otherwise, the inner expectation can be simplified as follows: $\mathbb{E}_{D|X}[[z(D)]_i] = 1 \times Q(D = i|X) + 0 \times (1 - Q(D = i|X))$, giving

$$\mathcal{L} = \mathbb{E}_X \left[ - \sum_{i=1}^{i=r} \log([f(X)]_i) Q(D = i|X) \right]$$

In order to learn a distribution, we constrain our search space to the subset of functions from $\mathbb{R}^p \to [0, 1]^r$ that project to a simplex $\Delta^{r-1}$. We now look to find the minimizer of the above equation from within this subset for each value of $X$. This corresponds to minimizing each term that contributes to the outer expectation over $X$, leading to the overall minimizer of the cross-entropy loss.

We formulate this objective along with the Lagrange constraint modeling the sum of components of $f(X)$ adding to 1. Note that we do not add the constraint of each component of $f(X)$ lying between 0 and 1, but it is easy to see that the resulting solution satisfies this constraint.

$$J = \min_{[f(X)]_1 \dots [f(X)]_r} - \sum_{i=1}^{i=r} \log([f(X)]_i) Q(D = i|X) + \lambda \left( \sum_{i=1}^{i=r} [f(X)]_i - 1 \right)$$

Setting partial derivative with respect to $[f(X)]_i$ to 0, we get $-\dfrac{Q(D = i|X)}{[f^\star(X)]_i} + \lambda = 0$ and $[f^\star(X)]_i = \frac{1}{\lambda} Q(D = i|X)$, where we use $f^\star$ to denote the minimizer.

The last piece is to solve for the Lagrange multiplier $\lambda$. This can be achieved by applying Karush-Kuhn-Tucker (KKT) conditions which suppose that the optimal solution lies on the constraint surface. This gives, $\sum_{i=1}^{i=r} [f^\star(X)]_i = 1$ which implies $\sum_{i=1}^{i=r} \frac{1}{\lambda} Q(D = i|X) = 1$. Since sum over domains $Q(D|X) = 1$, we get $\lambda = 1$.

Plugging this in, we finally get $[f^\star(X)]_i = Q(D = i|X)$. Thus, the optimal $f^*$ obtained by minimizing the cross entropy objective will in fact recover the oracle domain discriminator. As mentioned, it is clear that the resulting solution satisfies the required conditions of valid probability distributions.

$$\begin{bmatrix} 0.17 & 0.65 \\ 0.83 & 0.35 \end{bmatrix}$$
(a) $\alpha : 0.5, \kappa : 3$

$$\begin{bmatrix} 0.37 & 0.06 \\ 0.63 & 0.94 \end{bmatrix} \qquad \begin{bmatrix} 0.42 & 0.25 \\ 0.58 & 0.75 \end{bmatrix}$$
(b) $\alpha : 3, \kappa : 5$        (c) $\alpha : 10, \kappa : 7$

Figure 3: Example $\mathbf{Q}_{Y|D}$ matrices sampled for FieldGuide-2 with 2 classes and 2 domains. Each column represents the distribution across classes $p_d(y)$ for a given domain. At small $\alpha$, each $p_d(y)$ is likelier to be "sparse" (our definition is an informal one meaning not that there are many zero entries, but instead that the distribution is heavily concentrated in a few classes). At large $\alpha$, $p_d(y)$ tends toward a uniform distribution in which classes are represented evenly.

## D    Additional Experimental Details

Our code is available at `https://github.com/acmi-lab/Latent-Label-Shift-DDFA`. Here we present the full generation procedure for semisynthetic example problems, and discuss the parameters.

1. Choose a Dirichlet concentration parameter $\alpha > 0$, maximum condition number $\kappa \geqslant 1$ (with respect to 2-norm), and domain count $r \geqslant k$.
2. For each $y \in [k]$, sample $p_d(y) \sim \text{Dir}(\frac{\alpha}{k}\mathbf{1}_k)$.
3. Populate the matrix $\mathbf{Q}_{Y|D}$ with the computed $p_d(y)$s. If $\text{cond}(\mathbf{Q}_{Y|D}) > k$, return to step 2 and re-sample.
4. Distribute examples across domains according to $\mathbf{Q}_{Y|D}$, for each of train, test, and valid sets. This procedure entails creating a quota number of examples for each (class, domain) pair, and drawing datapoints without replacement to fill each quota. We must discard excess examples from some classes in the dataset due to class imbalance in the $\mathbf{Q}_{Y|D}$ matrix. Due to integral rounding, domains may be *slightly* imbalanced.
5. Conceal true class information and return $(x_i, d_i)$ pairs.

It is important to note the role of $\kappa$ and $\alpha$ in the above formulation. Although they are unknown parameters to the classification algorithm, they affect the sparsity of the $\mathbf{Q}_{Y|D}$ and difficulty of the problem. Small $\alpha$ encourages high sparsity in $p_d(y)$, and large $\alpha$ causes $p_d(y)$ to tend towards a uniform distribution. We observe an example of the effects of $\alpha$ in Fig. 3. $\kappa$ has a strong effect on the difficulty of the problem. Consider the case when $k = 2$. When $\kappa = 1$, the only potential $\mathbf{Q}_{Y|D}$ matrices are $\mathbf{I}_2$ up to row permutation (which means that domains and classes are exactly correlated, so the domain indicates the class and the problem is supervised). In the other limit, if $\kappa \to +\infty$, we may generate $\mathbf{Q}_{Y|D}$ matrices that are nearly singular, breaking needed assumptions for domain discriminator output to uniquely identify true class of anchor subdomains. $\kappa$ also helps control the class imbalance (if a row of $\mathbf{Q}_{Y|D}$ is small, indicating that the class is heavily under-represented across all domains, the condition number will increase).

### D.1   FieldGuide-2 and FieldGuide-28 Datasets

The dataset and description is available at `https://sites.google.com/view/fgvc6/competitions/butterflies-moths-2019`. From this data we create two datasets FieldGuide-2 and FieldGuide-28. For FieldGuide-28 we select the 28 classes which have 1000 datapoints in the training file. Since the test set provided in the website does not have annotations, we manually create a test set by sampling 200 datapoints from training file of each of the 28 classes. Therefore, we finally have 22400 training points and 5600 testing points. The FieldGuide-2 dataset is created by considering two classes from the created FieldGuide-28 dataset.

### D.2   Hyperparameters and Implementation Details: SCAN baseline

In all cases, we initialize the SCAN [72] network with the clustering head attached, sample data according to the $\mathbf{Q}_{Y|D}$ matrix, and predict classes. With the Hungarian algorithm, implemented in

[16, 74], we compute the highest true accuracy among any permutation of these labels (denoted "Test acc").

- **CIFAR-10 and CIFAR-20 Datasets [43]** We use ResNet-18 [35] backbone with weights trained by SCAN-loss and obtained from the SCAN repo `https://github.com/wvangansbeke/Unsupervised-Classification`. We use the same transforms present in the repo for test data.

- **ImageNet-50 Dataset [19]** We use ResNet-50 backbone with weights trained by SCAN-loss and obtained from the SCAN repo. We use the same transforms present in the repo for test data.

- **FieldGuide-2 and FieldGuide-28 Datasets** For each of the two datasets, we pretrain a different SCAN baseline network (including pretext and SCAN-loss steps) on all available data from the dataset. The backbone for each is ResNet-18. For training the pretext task, we use the same transform strategy used in the repo for CIFAR-10 data (with mean and std values as computed on the FieldGuide-28 dataset, and crop size 224). For training SCAN, we resize the smallest image dimension to 256, perform a random horizontal flip and random crop to size 224. We also normalize. For validation we resize smallest image dimension to 256, center crop to 224, and normalize.

  Hyperparameters used in training SCAN representations on both instances of the FieldGuide dataset were chosen as follows: starting with the recommended choices of hyperparameters for ImageNet (as was present in the SCAN repo), we made minimal changes to these only to avoid model degeneracy (training loss collapse).

### D.3 Hyperparameters and Implementation Details: DDFA (RI)

This is the DDFA procedure with random initialization of the domain discriminator. The bulk of this procedure is described in Section 6, but for completeness we reiterate here.

We train ResNet-50 [35] (with random initialization and added dropout) based on the implementation from `https://github.com/kuangliu/pytorch-cifar` on images $x_i$ with domain indices $d_i$ as the label, choose best iteration by valid loss, pass all training and validation data through $\hat{f}$, and cluster pushforward predictions $\hat{f}(x_i)$ into $m \geqslant k$ clusters with Faiss K-Means [42]. We compute the $\hat{\mathbf{Q}}_{c(X)|D}$ matrix and run NMF to obtain $\hat{\mathbf{Q}}_{c(X)|Y}$, $\hat{\mathbf{Q}}_{Y|D}$. To make columns sum to 1, we normalize columns of $\hat{\mathbf{Q}}_{c(X)|Y}$, multiply each column's normalization coefficient over the corresponding row of $\hat{\mathbf{Q}}_{Y|D}$ (to preserve correctness of the decomposition), and then normalize columns of $\hat{\mathbf{Q}}_{Y|D}$.

Some NMF algorithms only output solutions satisfying the anchor word property [4, 44, 30]. We found the strict requirement of an exact anchor word solution to lead to low noise tolerance. We therefore use the Sklearn implementation of standard NMF [15, 69, 59].

We predict class labels with Algorithm 2. With the Hungarian algorithm, implemented in [16, 74], we compute the highest true accuracy among any permutation of these labels (denoted "Test acc"). With the same permutation, we reorder rows of $\hat{Q}_{Y|D}$, then compute the average absolute difference between corresponding entries of $\hat{\mathbf{Q}}_{Y|D}$ and $\mathbf{Q}_{Y|D}$ (denoted "$\mathbf{Q}_{Y|D}$ err").

In order to make hyperparameter choices for final experiments, such as the choice of the NMF solver, clustering algorithm, and learning rate, we primarily consulted CIFAR-10 and CINIC-10 (similar to an extension of CIFAR-10) [17] final test task accuracy, and validation loss on other datasets (likely leading to an overfitting of our hyperparameter choices to CIFAR-10 and associated tasks). We applied the intuitions developed on these datasets when choosing hyperparameters for other datasets, instead of performing a test accuracy-driven sweep for each other dataset. Final runs used the following fixed hyperparameters:

**Common Hyperparameters**

- Hardware: A single NVIDIA RTX A6000 GPU was used for each experiment (on this hardware, trial length varies < 1 hour to 24 hours, depending on the dataset).
- Architecture: ResNet-50 with added dropout
- Faiss KMeans number of iterations (niter): 100

- Faiss Kmeans number of clustering redos (nredo): 5
- Learning Rate: 0.001
- Learning Rate Decay: Exponential, parameter 0.97
- SKlearn NMF initialization: random

**Dataset-Specific Hyperparameters**

- **CIFAR-10 Dataset** Training Epochs: 100. Number of Clusters ($m$): 30
- **CIFAR-20 Dataset** Training Epochs: 100. Number of Clusters ($m$): 60
- **ImageNet-50 Dataset** DDFA (RI) was not evaluated for this dataset due to poor performance of the domain discriminator without an appropriate pre-seed in early trials.
- **FieldGuide-2 Dataset** Training Epochs: 100. Number of Clusters ($m$): 10
- **FieldGuide-28 Dataset** DDFA (RI) was not evaluated for this dataset due to poor performance of the domain discriminator without an appropriate pre-seed in early trials.

## D.4 Hyperparameters and Implementation Details: DDFA (SI) and DDFA (SPI)

This is the DDFA procedure with SCAN initialization of the domain discriminator. DDFA (SI) uses the SCAN pretext + SCAN loss pretraining steps, while DDFA (SPI) uses only the SCAN pretext step.

The procedure is identical to the standard DDFA procedure, except that SCAN [72] pre-trained weights or SCAN [72] contrastive pre-text weights are used to initialize the domain discriminator before it is fine-tuned on the domain discrimination task. Hyperparameters used also differ. When SCAN pretrained weights are available, we use those. When they are not, we train SCAN ourselves.

Like SCAN (RI), we used CIFAR-10 and CINIC-10 final test accuracy to choose hyperparameters and make algorithm decisions. For other datasets, we consulted only validation domain discrimination loss. One exception to this rule was that preliminary low final DDFA (SI) performance on FieldGuide suggested that we should focus on instead evaluating DDFA (SPI) to avoid allowing the SCAN failure mode to negatively affect the domain discriminator pretrain representation. Final evaluation runs used the following fixed hyperparameters:

**Common Hyperparameters**

- Hardware: A single NVIDIA RTX A6000 GPU was used for each experiment (on this hardware, trial length varies < 1 hour to 24 hours, depending on the dataset).
- Faiss KMeans number of iterations (niter): 100
- Faiss Kmeans number of clustering redos (nredo): 5
- Learning Rate: 0.00001
- Learning Rate Decay: Exponential, parameter 0.97
- SKlearn NMF initialization: random

**Dataset-Specific Hyperparameters**

- **CIFAR-10 Dataset**
  Architecture: ResNet-18
  Pre-seed: Weights trained with SCAN pretext and SCAN-loss on entirety of CIFAR-10 (from SCAN repo).
  Training Epochs: 25
  Number of Clusters ($m$): 10
  Transforms used: Same as SCAN repo.

- **CIFAR-20 Dataset**
  Architecture: ResNet-18
  Pre-seed: Weights trained with SCAN pretext and SCAN-loss on entirety of CIFAR-20 (from SCAN repo).

Training Epochs: 25

Number of Clusters ($m$): 20

Transforms used: Same as SCAN repo.

- **ImageNet-50 Dataset**

  Architecture: ResNet-50

  Pre-seed: Weights trained with SCAN pretext and SCAN-loss on entirety of ImageNet-50 (from SCAN repo).

  Training Epochs: 25

  Number of Clusters ($m$): 50

  Transforms used: Same as SCAN repo.

- **FieldGuide-2 Dataset**

  Architecture: ResNet-18

  Pre-seed: Weights trained with SCAN pretext on entirety of FieldGuide-2 (trained by us).

  Training Epochs: 30

  Number of Clusters ($m$): 2

  Transforms used for pretext: Same strategy as CIFAR-10 in SCAN repo with appropriate mean, std, and crop size 224.

  Transform used for SCAN: Resize to 256, Random horizontal flip, Random crop to 224, normalize

  Learning rate used for SCAN: 0.001 (other hyperparameters were same as in SCAN repo for CIFAR-10)

- **FieldGuide-28 Dataset**

  Architecture: ResNet-18

  Pre-seed: Weights trained with SCAN pretext on entirety of FieldGuide-28 (trained by us).

  Training Epochs: 60

  Number of Clusters ($m$): 28

  Transforms used for pretext: Same strategy as CIFAR-10 in SCAN repo with appropriate mean, std, and crop size 224.

  Transform used for SCAN: Resize to 256, Random horizontal flip, Random crop to 224, normalize

  Learning rate used for SCAN: 0.01 (other hyperparameters were same as in SCAN repo for CIFAR-10)

# E    Additional Experimental Results

Here we present additional experimental results. We also investigated evaluations on the Waterbirds dataset, but although DDFA showed some reasonable results, we did not find SCAN hyperparameters that lead to a successful SCAN baseline. Accordingly, we do not include these results as they do not present a reasonable comparison.

Table 2: *Results on CIFAR-10.* Each entry is produced with the averaged result of 3 different random seeds, formatted as mean $\pm$ standard deviation. With DDFA (RI) we refer to DDFA with randomly initialized backbone. With DDFA (SI) we refer to DDFA's backbone initialized with SCAN. Note that in DDFA (SI), we do not leverage SCAN for clustering. $\alpha$ is the Dirichlet parameter used for generating label marginals in each domain, $\kappa$ is the maximum allowed condition number of the generated $\mathbf{Q}_{Y|D}$ matrix, $r$ is number of domains. "Test acc" is classification accuracy, under the best permutation of the recovered classes, and "$\mathbf{Q}_{Y|D}$ err" is the average entry-wise absolute error in the recovered $\mathbf{Q}_{Y|D}$.

| r | Approaches | $\alpha: 0.5,\ \kappa: 4$ | | $\alpha: 3,\ \kappa: 4$ | | $\alpha: 10,\ \kappa: 8$ | |
|---|---|---|---|---|---|---|---|
| | | Test acc | $\mathbf{Q}_{Y|D}$ err | Test acc | $\mathbf{Q}_{Y|D}$ err | Test acc | $\mathbf{Q}_{Y|D}$ err |
| 10 | SCAN | $0.808 \pm 0.007$ | $0.066 \pm 0.002$ | $\mathbf{0.823} \pm 0.007$ | $0.050 \pm 0.003$ | $\mathbf{0.815} \pm 0.005$ | $\mathbf{0.036} \pm 0.002$ |
| | DDFA (RI) | $0.759 \pm 0.035$ | $0.033 \pm 0.003$ | $0.564 \pm 0.042$ | $0.054 \pm 0.003$ | $0.296 \pm 0.027$ | $0.075 \pm 0.005$ |
| | DDFA (SI) | $\mathbf{0.867} \pm 0.089$ | $\mathbf{0.029} \pm 0.014$ | $0.728 \pm 0.068$ | $\mathbf{0.047} \pm 0.008$ | $0.584 \pm 0.022$ | $0.058 \pm 0.002$ |
| 15 | SCAN | $0.823 \pm 0.021$ | $0.061 \pm 0.004$ | $0.817 \pm 0.017$ | $0.052 \pm 0.003$ | $0.821 \pm 0.007$ | $\mathbf{0.036} \pm 0.000$ |
| | DDFA (RI) | $0.750 \pm 0.057$ | $0.040 \pm 0.009$ | $0.538 \pm 0.072$ | $0.058 \pm 0.010$ | $0.329 \pm 0.034$ | $0.070 \pm 0.011$ |
| | DDFA (SI) | $\mathbf{0.921} \pm 0.040$ | $\mathbf{0.023} \pm 0.010$ | $\mathbf{0.849} \pm 0.078$ | $\mathbf{0.026} \pm 0.013$ | $0.709 \pm 0.053$ | $0.038 \pm 0.007$ |
| 20 | SCAN | $0.813 \pm 0.017$ | $0.064 \pm 0.002$ | $0.814 \pm 0.008$ | $0.049 \pm 0.004$ | $\mathbf{0.804} \pm 0.012$ | $0.035 \pm 0.001$ |
| | DDFA (RI) | $0.722 \pm 0.094$ | $0.034 \pm 0.013$ | $0.510 \pm 0.071$ | $0.060 \pm 0.014$ | $0.251 \pm 0.049$ | $0.071 \pm 0.013$ |
| | DDFA (SI) | $\mathbf{0.898} \pm 0.010$ | $\mathbf{0.025} \pm 0.001$ | $\mathbf{0.901} \pm 0.025$ | $\mathbf{0.016} \pm 0.005$ | $0.765 \pm 0.038$ | $\mathbf{0.034} \pm 0.004$ |
| 25 | SCAN | $0.800 \pm 0.015$ | $0.067 \pm 0.002$ | $0.821 \pm 0.008$ | $0.047 \pm 0.002$ | $\mathbf{0.814} \pm 0.017$ | $0.036 \pm 0.001$ |
| | DDFA (RI) | $0.683 \pm 0.093$ | $0.057 \pm 0.025$ | $0.527 \pm 0.068$ | $0.050 \pm 0.013$ | $0.310 \pm 0.066$ | $0.065 \pm 0.010$ |
| | DDFA (SI) | $\mathbf{0.968} \pm 0.004$ | $\mathbf{0.018} \pm 0.004$ | $\mathbf{0.918} \pm 0.006$ | $\mathbf{0.018} \pm 0.004$ | $0.775 \pm 0.048$ | $\mathbf{0.029} \pm 0.006$ |

Table 3: *Full results on CIFAR-20.* Each entry is produced with the averaged result of 3 different random seeds, formatted as mean $\pm$ standard deviation. With DDFA (RI) we refer to DDFA with randomly initialized backbone. With DDFA (SI) we refer to DDFA's backbone initialized with SCAN. Note that in DDFA (SI), we do not leverage SCAN for clustering. $\alpha$ is the Dirichlet parameter used for generating label marginals in each domain, $\kappa$ is the maximum allowed condition number of the generated $\mathbf{Q}_{Y|D}$ matrix, $r$ is number of domains.

| r | Approaches | $\alpha: 0.5,\ \kappa: 8$ | | $\alpha: 3,\ \kappa: 12$ | | $\alpha: 10,\ \kappa: 20$ | |
|---|---|---|---|---|---|---|---|
| | | Test acc | $\mathbf{Q}_{Y|D}$ err | Test acc | $\mathbf{Q}_{Y|D}$ err | Test acc | $\mathbf{Q}_{Y|D}$ err |
| 20 | SCAN | $0.454 \pm 0.016$ | $0.059 \pm 0.002$ | $0.421 \pm 0.010$ | $0.051 \pm 0.001$ | $\mathbf{0.436} \pm 0.009$ | $0.038 \pm 0.001$ |
| | DDFA (RI) | $0.520 \pm 0.064$ | $0.041 \pm 0.005$ | $0.357 \pm 0.020$ | $0.043 \pm 0.003$ | $0.187 \pm 0.008$ | $0.051 \pm 0.003$ |
| | DDFA (SI) | $\mathbf{0.852} \pm 0.015$ | $\mathbf{0.015} \pm 0.000$ | $\mathbf{0.548} \pm 0.062$ | $\mathbf{0.026} \pm 0.003$ | $0.354 \pm 0.096$ | $\mathbf{0.036} \pm 0.007$ |
| 25 | SCAN | $0.458 \pm 0.055$ | $0.059 \pm 0.004$ | $0.455 \pm 0.011$ | $0.048 \pm 0.001$ | $0.440 \pm 0.023$ | $0.037 \pm 0.001$ |
| | DDFA (RI) | $0.525 \pm 0.033$ | $0.042 \pm 0.006$ | $0.310 \pm 0.020$ | $0.051 \pm 0.006$ | $0.182 \pm 0.002$ | $0.053 \pm 0.003$ |
| | DDFA (SI) | $\mathbf{0.819} \pm 0.013$ | $\mathbf{0.021} \pm 0.002$ | $\mathbf{0.707} \pm 0.022$ | $\mathbf{0.022} \pm 0.004$ | $\mathbf{0.502} \pm 0.024$ | $\mathbf{0.030} \pm 0.002$ |
| 30 | SCAN | $0.456 \pm 0.012$ | $0.059 \pm 0.001$ | $0.441 \pm 0.023$ | $0.050 \pm 0.001$ | $0.437 \pm 0.010$ | $0.037 \pm 0.001$ |
| | DDFA (RI) | $0.506 \pm 0.104$ | $0.045 \pm 0.009$ | $0.256 \pm 0.007$ | $0.058 \pm 0.005$ | $0.088 \pm 0.016$ | $0.076 \pm 0.006$ |
| | DDFA (SI) | $\mathbf{0.845} \pm 0.041$ | $\mathbf{0.020} \pm 0.006$ | $\mathbf{0.688} \pm 0.023$ | $\mathbf{0.027} \pm 0.003$ | $\mathbf{0.531} \pm 0.034$ | $\mathbf{0.029} \pm 0.002$ |

Table 4: *Results on ImageNet-50*. Each entry is produced with the averaged result of 3 different random seeds, formatted as mean $\pm$ standard deviation. With DDFA (SI) we refer to DDFA's backbone initialized with SCAN. Note that in DDFA (SI), we do not leverage SCAN for clustering. $\alpha$ is the Dirichlet parameter used for generating label marginals in each domain, $\kappa$ is the maximum allowed condition number of the generated $\mathbf{Q}_{Y|D}$ matrix, $r$ is number of domains. "Test acc" is classification accuracy, under the best permutation of the recovered classes, and "$\mathbf{Q}_{Y|D}$ err" is the average entry-wise absolute error in the recovered $\mathbf{Q}_{Y|D}$.

| r | Approaches | $\alpha: 0.5, \kappa: 200$ | | $\alpha: 3, \kappa: 205$ | | $\alpha: 10, \kappa: 210$ | |
| | | Test acc | $\mathbf{Q}_{Y|D}$ err | Test acc | $\mathbf{Q}_{Y|D}$ err | Test acc | $\mathbf{Q}_{Y|D}$ err |
|---|---|---|---|---|---|---|---|
| 50 | SCAN | **0.751** $\pm$ 0.058 | **0.011** $\pm$ 0.002 | **0.753** $\pm$ 0.036 | **0.010** $\pm$ 0.001 | **0.738** $\pm$ 0.010 | **0.009** $\pm$ 0.000 |
| | DDFA (SI) | 0.745 $\pm$ 0.021 | **0.011** $\pm$ 0.002 | 0.569 $\pm$ 0.059 | 0.016 $\pm$ 0.002 | 0.380 $\pm$ 0.126 | 0.021 $\pm$ 0.002 |
| 60 | SCAN | 0.752 $\pm$ 0.028 | 0.011 $\pm$ 0.001 | **0.765** $\pm$ 0.030 | **0.010** $\pm$ 0.001 | **0.743** $\pm$ 0.008 | **0.008** $\pm$ 0.000 |
| | DDFA (SI) | **0.790** $\pm$ 0.057 | **0.009** $\pm$ 0.002 | 0.693 $\pm$ 0.032 | 0.013 $\pm$ 0.002 | 0.585 $\pm$ 0.028 | 0.017 $\pm$ 0.002 |

Table 5: *Full results on FieldGuide-2*. Each entry is produced with the averaged result of 3 different random seeds, formatted as mean $\pm$ standard deviation. With DDFA (RI) we refer to DDFA with randomly initialized backbone. With DDFA (SPI) we refer to DDFA initialized with pretext training adopted by SCAN. Note that in DDFA (SPI), we do not leverage SCAN for clustering. $\alpha$ is the Dirichlet parameter used for generating label marginals in each domain, $\kappa$ is the maximum allowed condition number of the generated $\mathbf{Q}_{Y|D}$ matrix, $r$ is number of domains.

| r | Approaches | $\alpha: 0.5, \kappa: 3$ | | $\alpha: 3, \kappa: 5$ | | $\alpha: 10, \kappa: 7$ | |
| | | Test acc | $\mathbf{Q}_{Y|D}$ err | Test acc | $\mathbf{Q}_{Y|D}$ err | Test acc | $\mathbf{Q}_{Y|D}$ err |
|---|---|---|---|---|---|---|---|
| 2 | SCAN | 0.588 $\pm$ 0.007 | 0.382 $\pm$ 0.059 | 0.580 $\pm$ 0.008 | **0.158** $\pm$ 0.094 | 0.597 $\pm$ 0.011 | **0.139** $\pm$ 0.022 |
| | DDFA (SPI) | **0.950** $\pm$ 0.049 | **0.089** $\pm$ 0.039 | **0.685** $\pm$ 0.165 | 0.229 $\pm$ 0.079 | **0.658** $\pm$ 0.068 | 0.290 $\pm$ 0.046 |
| 3 | SCAN | 0.604 $\pm$ 0.027 | 0.324 $\pm$ 0.022 | 0.582 $\pm$ 0.007 | **0.200** $\pm$ 0.085 | **0.587** $\pm$ 0.006 | **0.113** $\pm$ 0.056 |
| | DDFA (SPI) | **0.915** $\pm$ 0.042 | **0.125** $\pm$ 0.072 | **0.763** $\pm$ 0.073 | 0.219 $\pm$ 0.125 | 0.571 $\pm$ 0.102 | 0.265 $\pm$ 0.095 |
| 5 | SCAN | 0.582 $\pm$ 0.015 | 0.320 $\pm$ 0.054 | 0.590 $\pm$ 0.016 | **0.219** $\pm$ 0.090 | 0.585 $\pm$ 0.010 | **0.139** $\pm$ 0.036 |
| | DDFA (SPI) | **0.863** $\pm$ 0.087 | **0.204** $\pm$ 0.129 | **0.794** $\pm$ 0.077 | 0.235 $\pm$ 0.084 | **0.588** $\pm$ 0.092 | 0.324 $\pm$ 0.057 |
| 7 | SCAN | 0.573 $\pm$ 0.028 | 0.306 $\pm$ 0.010 | 0.594 $\pm$ 0.028 | **0.183** $\pm$ 0.010 | 0.580 $\pm$ 0.007 | **0.102** $\pm$ 0.006 |
| | DDFA (SPI) | **0.911** $\pm$ 0.016 | **0.087** $\pm$ 0.082 | **0.696** $\pm$ 0.123 | 0.284 $\pm$ 0.108 | **0.592** $\pm$ 0.056 | 0.325 $\pm$ 0.047 |
| 10 | SCAN | 0.597 $\pm$ 0.014 | 0.304 $\pm$ 0.060 | 0.590 $\pm$ 0.003 | 0.155 $\pm$ 0.029 | **0.589** $\pm$ 0.004 | **0.099** $\pm$ 0.009 |
| | DDFA (SPI) | **0.844** $\pm$ 0.075 | **0.225** $\pm$ 0.081 | **0.756** $\pm$ 0.049 | **0.116** $\pm$ 0.015 | 0.587 $\pm$ 0.063 | 0.157 $\pm$ 0.091 |

Table 6: *Results on FieldGuide-28*. Each entry is produced with the averaged result of 3 different random seeds, formatted as mean $\pm$ standard deviation. With DDFA (SPI) we refer to DDFA initialized with pretext training adopted by SCAN. Note that in DDFA (SPI), we do not leverage SCAN for clustering. $\alpha$ is the Dirichlet parameter used for generating label marginals in each domain, $\kappa$ is the maximum allowed condition number of the generated $\mathbf{Q}_{Y|D}$ matrix, $r$ is number of domains. "Test acc" is classification accuracy, under the best permutation of the recovered classes, and "$\mathbf{Q}_{Y|D}$ err" is the average entry-wise absolute error in the recovered $\mathbf{Q}_{Y|D}$.

| r | Approaches | $\alpha : 0.5,\ \kappa : 12$ | | $\alpha : 3,\ \kappa : 20$ | | $\alpha : 10,\ \kappa : 28$ | |
|---|---|---|---|---|---|---|---|
| | | Test acc | $\mathbf{Q}_{Y|D}$ err | Test acc | $\mathbf{Q}_{Y|D}$ err | Test acc | $\mathbf{Q}_{Y|D}$ err |
| 28 | SCAN | $0.254 \pm 0.002$ | $0.050 \pm 0.000$ | $0.251 \pm 0.013$ | $0.043 \pm 0.001$ | $0.251 \pm 0.009$ | $\mathbf{0.032} \pm 0.001$ |
| | DDFA (SPI) | $\mathbf{0.549} \pm 0.051$ | $\mathbf{0.032} \pm 0.002$ | $\mathbf{0.363} \pm 0.056$ | $\mathbf{0.036} \pm 0.002$ | $\mathbf{0.295} \pm 0.028$ | $0.036 \pm 0.002$ |
| 37 | SCAN | $0.239 \pm 0.013$ | $0.053 \pm 0.001$ | $0.265 \pm 0.020$ | $0.041 \pm 0.001$ | $0.250 \pm 0.014$ | $\mathbf{0.031} \pm 0.001$ |
| | DDFA (SPI) | $\mathbf{0.705} \pm 0.063$ | $\mathbf{0.031} \pm 0.003$ | $\mathbf{0.551} \pm 0.036$ | $\mathbf{0.031} \pm 0.001$ | $\mathbf{0.348} \pm 0.033$ | $0.034 \pm 0.002$ |
| 42 | SCAN | $0.277 \pm 0.026$ | $0.050 \pm 0.001$ | $0.264 \pm 0.010$ | $0.042 \pm 0.000$ | $0.258 \pm 0.012$ | $\mathbf{0.031} \pm 0.001$ |
| | DDFA (SPI) | $\mathbf{0.684} \pm 0.030$ | $\mathbf{0.033} \pm 0.001$ | $\mathbf{0.492} \pm 0.062$ | $\mathbf{0.036} \pm 0.003$ | $\mathbf{0.388} \pm 0.042$ | $0.034 \pm 0.002$ |
| 47 | SCAN | $0.242 \pm 0.013$ | $0.051 \pm 0.001$ | $0.271 \pm 0.005$ | $0.041 \pm 0.001$ | $0.243 \pm 0.004$ | $\mathbf{0.032} \pm 0.000$ |
| | DDFA (SPI) | $\mathbf{0.747} \pm 0.041$ | $\mathbf{0.030} \pm 0.005$ | $\mathbf{0.506} \pm 0.070$ | $\mathbf{0.036} \pm 0.003$ | $\mathbf{0.354} \pm 0.046$ | $0.036 \pm 0.003$ |

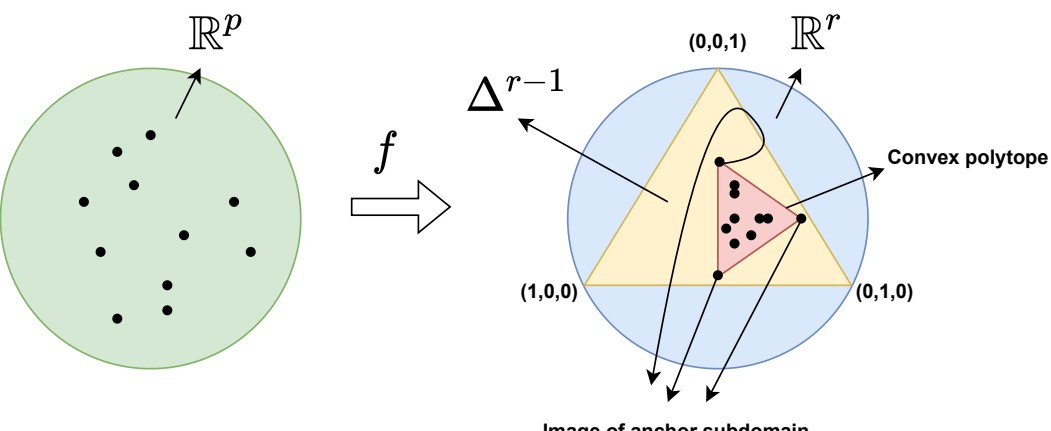

Figure 4: This figure illustrates the case with 3 domains and 3 classes. The oracle domain discriminator maps points from a high-dimensional input space to a $k = 3$ vertex convex polytope (shaded red) embedded in $\Delta^{r-1}, r = 3$ (shaded yellow). The anchor subdomains map to the vertices of this polytope.

# F   Discussion of Convex Polytope Geometry

The geometric properties of topic modeling for finite, discrete random variables has been explored in depth in related works ([38, 22, 14]). The observation that columns in $\mathbf{Q}_{X|D}$ are convex combinations of columns in $\mathbf{Q}_{X|Y}$ leads to a perspective on identification of the matrix decomposition as identification of the convex polytope in $\mathbb{R}^m$ which encloses all of the columns of $\mathbf{Q}_{X|D}$ (the corners of which correspond to columns of $\mathbf{Q}_{X|Y}$ under certain identifiability conditions).

Here, we briefly discuss an interesting but somewhat different application of convex polytope geometry. Instead of a convex polytope in $\mathbb{R}^m$ with corners as columns of $\mathbf{Q}_{X|Y}$, we concern ourselves with the convex polytope in $\mathbb{R}^r$ with corners as columns in $\mathbf{Q}_{D|Y}$, which must enclose all values taken by the oracle domain discriminator $f(x)$ for $x \in \mathcal{X}, q(x) > 0$.

Let us assume that Assumptions A.1–A.4 are satisfied. We recall the oracle domain discriminator $f$ which is defined such that $[f(x)]_d = q(d|X = x)$. Let $x \in \mathcal{X} = \mathbb{R}^p$. Now, since the $r$ values $q(d|X = x)$ for $d \in \{1, 2, ..., r\}$ together constitute a categorical distribution, each of these $r$ values lie between 0 and 1, and also their sum adds to 1. Therefore the vector $f(x)$ lies on the simplex $\Delta^{r-1}$. We now express $f(x)$ as a convex combination of the $k$ columns of $\mathbf{Q}_{D|Y}$. We denote these column vectors $\mathbf{Q}_{D|Y}[:, y]$ for each $y \in \mathcal{Y} = [k]$. Note that each such vector also lies in the $\Delta^{r-1}$ simplex.

As an intermediate step in the proof of Lemma 3 given in App. A, we showed that each $f(x)$ is a linear combination of these columns of $\mathbf{Q}_{D|Y}$ with coefficients $q(y|X = x)$ for all $y \in \mathcal{Y}$. That is, we can rewrite $f(x) = \mathbf{Q}_{D|Y} \left[ Q(Y = 1|X = x) \dots Q(Y = k|X = x) \right]^\top$

Since the coefficients in the linear combination are probabilities which, taken together, form a categorical distribution, they lie between 0 and 1 and sum to 1. Thus, for all $x \in \mathcal{X}$ with $q(x) > 0$, $f(x)$ can be expressed as a *convex* combination of the columns of $\mathbf{Q}_{D|Y}$. Therefore, for any $x$ with $q(x) > 0$, $f(x)$ lies inside the $k-$vertex convex polytope with corners as the columns of $\mathbf{Q}_{D|Y}$ (which are linearly independent by Lemma 6). This polytope is embedded in $\Delta^{r-1}$.

Now consider $x$ in an anchor sub-domain, that is $x \in A_y$ for some $y \in \mathcal{Y}$. We know that if $q(x) > 0$, $q(y|X = x) = 1$, $q(y'|X = x) = 0$ for all $y' \neq y$ (Lemma 5). Since the $q(y|X = x)$ are now one-hot, we have that $f(x) = \mathbf{Q}_{D|Y}[:, y]$ for $x \in A_y$. In words, this means that $f(A_y)$ is precisely the $y$th column of $\mathbf{Q}_{D|Y}$. It follows that the domain discriminator maps each of the $k$ anchor sub-domains exactly to a unique vertex of the polytope. The situation is described in Fig. 4.

We could now recover the columns of $\mathbf{Q}_{D|Y}$, up to permutation, with the following procedure:

1. Push all $x \in \mathcal{X}$ through $f$.
2. Find the minimum volume convex polytope that contains the resulting density of points on the simplex. The vectors that compose the vertices of this polytope are the columns of $\mathbf{Q}_{D|Y}$, up to permutation.

Note that from Assumption A.4, we are guaranteed to have a region of the input space with at least $\epsilon > 0$ mass that gets mapped to each of the vertices when carrying out step (i). Therefore, our discovered minimum volume polytope must enclose all of these vertices. Since no mass will exist outside of the true polytope, requiring a minimum volume polytope will ensure that the recovered polytope fits the true polytope's vertices precisely (as any extraneous volume outside of the true polytope must be eliminated). Then step (ii) recovers $\mathbf{Q}_{D|Y}$, up to permutation of columns. Having recovered $\mathbf{Q}_{D|Y}$, we can use Lemmas 1 and 2 to recover $q(y|x, d)$.

This procedure is a geometric alternative to the clustering approach outlined in Algorithm 1. In practice, fitting a convex hull around the outputs of a noisy, non-oracle estimated domain discriminator may be computationally expensive, and noise may lead this sensitive procedure to fail to recover the true vertices.

## G   Ablation Study on Number of Clusters

We conduct an ablation on the choice of $m$, the parameter indicating how many clusters to find in the $q(d|x)$ space. We use the CIFAR-20 dataset with 20 domains and employ DDFA (SI) and SCAN models, following the same hyperparameters as outlined in App. D, except for modifying the choice of $m$ for DDFA (SI). Results are obtained as the average of 3 random seeds.

The number of true classes is 20 in CIFAR-20. As seen in Fig. 5, when $m$ is chosen to be 10, violating the typical constraint that $m \geqslant k$, we can still solve for the solution, but we get poor performance, seeing a drop in accuracy as much as 20% from a better-chosen value of $m$. Choosing $m$ directly equal to or larger (up to 50) than $k$ provides the best performance, with a slope-off in performance at very large $m$, although the effect is very slight for the alpha = 0.5 setting.

The trend is roughly mirrored in Fig. 6, which shows how the reconstruction error changes over the same variation in $m$. Under all settings, using $m = 10 < k$ clusters provides a poor reconstruction, while the best reconstruction is found with $m$ equal to $k$. Performance degrades as $m$ grows very large, although the effect is very slight for the alpha = 0.5 setting.

Intuitively, these results show that breaking the $m \geqslant k$ condition not only violates the theoretical identifiability, but also leads to poorer empirical performance; choosing very large $m$ can also lead to degraded performance, likely due to propagation of inaccuracies in the finite-sample estimation of the $\widehat{Q}_{c(X)|D}$ matrix.

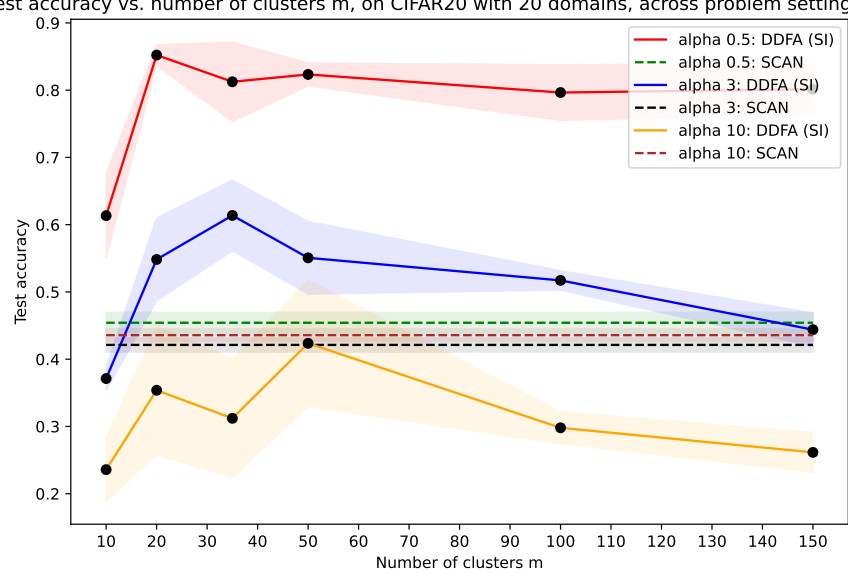

Figure 5: Test accuracy of DDFA (SI) approach and SCAN baseline on CIFAR20 with 20 domains, while modifying the number of clusters $m$ for DDFA (SI). The choice of $\alpha$ roughly modifies the difficulty of the problem, where small $\alpha$ is easier. We note that typically we require choice of $m \geqslant k$. We portray one datapoint where this constraint is violated, and $m = 10$. Black dots indicate tested values of $m$, and lines are plotted only to show the trend. Larger accuracy is better. Mean $\pm$ std reported.

# H  Ablation Study with a Naïve Feature Space

One might ask whether the semantic meaning of the domain discrimination space is necessary in DDFA; might we exchange the domain discrimination step in Algorithm 1 for a naive step in which we simply pass the input through an arbitrary feature extractor and then proceed to clustering in this space?

The first remark we make is that the domain discriminator does not purely provide a clustering representation; its semantic meaning is also important for the computation of the final domain-adjusted $p_d(y|x)$ prediction, as a reliable estimate of $q(d|x)$, in conjunction with the estimate of the $Q_{Y|D}$ matrix, allow us to estimate $p_d(y|x)$ via Algorithm 2. Without this semantic meaning, our class prediction can be based only on a coarse prediction at the level of the *cluster*, not the individual datapoint.

However, if we are still determined to use an alternate representation, it is indeed possible to do so. We illustrate a variant of DDFA using a naïve representation in Algorithms 3 and 4, and then evaluate this procedure on CIFAR-20 as an ablative study on DDFA. We compare naïve results with standard DDFA results, and with a traditional SCAN baseline, in Table 7.

**Naïve Representation Variant of DDFA**    The only major changes from the original DDFA for Algorithm 3 are the removal of the need to train any $\hat{f}$ domain discriminator, the use of the arbitrary representation space $\phi$ before clustering, and the reliance on the output of the clustering discretization function $c$ as well as $\widehat{\mathbf{Q}}_{c(X)|Y}$ (which are both discarded in the original procedure). We need $c$ and $\widehat{\mathbf{Q}}_{c(X)|Y}$ because in Algorithm 4 we will use them for domain-adjusted class prediction.

Algorithm 4 includes significant changes from the DDFA procedure. Since we do not have the estimate of $q(d|x)$ to use, we cannot directly reason about how different locations in the representation space induced by $\phi$ correspond to different probabilities of class labels. However, because we have $\widehat{\mathbf{Q}}_{c(X)|Y}$ and $\widehat{\mathbf{Q}}_{Y|D}$, two outputs of the NMF decomposition in Algorithm 3, we can calculate a

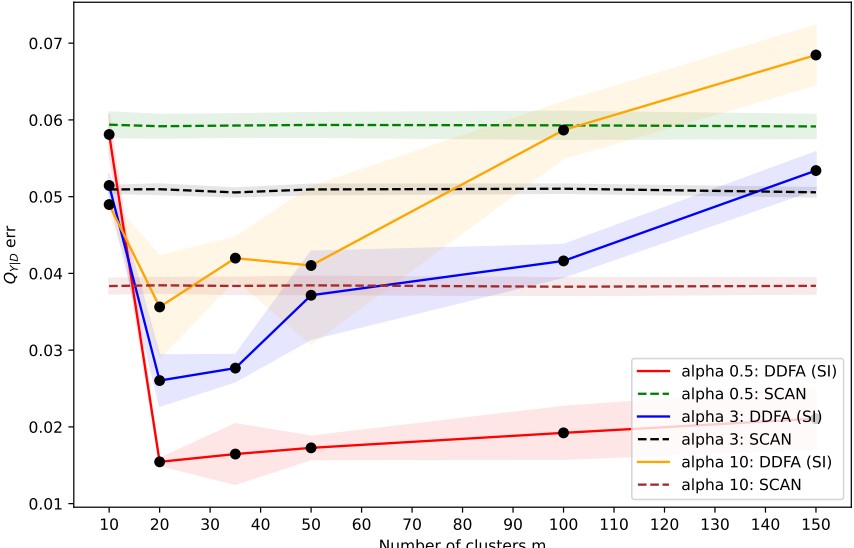

Figure 6: Element-wise average absolute $Q_{Y|D}$ reconstruction error of DDFA (SI) approach and SCAN baseline on CIFAR-20 with 20 domains, while modifying the number of clusters $m$ for DDFA (SI). The choice of $\alpha$ roughly modifies the difficulty of the problem, where small $\alpha$ is easier. We note that typically we require choice of $m \geqslant k$. We portray one datapoint where this constraint is violated, and $m = 10$. Black dots indicate tested values of $m$, and lines are plotted only to show the trend. Smaller error is better. Mean $\pm$ std reported.

---

**Algorithm 3** DDFA (Naïve) Training

**input** $k \geqslant 1, r \geqslant k, \{(x_i, d_i)\}_{i \in [n]} \sim q(x, d)$, A naive representation function $\phi$ from $\mathbb{R}^p \to \mathbb{R}^r$
1: Push all $\{x_i\}_{i \in [n]}$ through $\phi$.
2: Train clustering algorithm on the n points $\{\phi(x_i)\}_{i \in [n]}$, obtain $m$ clusters.
3: $c(x_i) \leftarrow$ Cluster id of $\phi(x_i)$
4: $\widehat{q}(c(X) = a | D = b) \leftarrow \dfrac{\sum_{i \in [n]} \mathbb{I}[c(x_i) = a, \, d_i = b]}{\sum_{j \in [n]} \mathbb{I}[d_j = b]}$
5: Populate $\widehat{\mathbf{Q}}_{c(X)|D}$ as $[\widehat{\mathbf{Q}}_{c(X)|D}]_{a,b} \leftarrow \widehat{q}(c(X) = a | D = b)$
6: $\widehat{\mathbf{Q}}_{c(X)|Y}, \widehat{\mathbf{Q}}_{Y|D} \leftarrow$ NMF $(\widehat{\mathbf{Q}}_{c(X)|D})$
**output** $\widehat{\mathbf{Q}}_{c(X)|Y}, \widehat{\mathbf{Q}}_{Y|D}$, clustering discretization function $c$

---

coarse prediction over labels $y$ for each cluster, and then assign the same prediction to each point in that cluster. To obtain the closed-form for this coarse prediction $\widehat{p}_d(y|c(x))$ used in Algorithm 4, we use the following derivation:

$$
\begin{aligned}
\widehat{p}_d(y|c(x)) = \widehat{q}(y|d, c(x)) &= \frac{\widehat{q}(c(x)|y, d)\widehat{q}(y, d)}{\widehat{q}(d, c(x))} \\
&= \frac{\widehat{q}(c(x)|y, d)\widehat{q}(y, d)}{\sum\limits_{y'' \in \mathcal{Y}} \widehat{q}(c(x)|y'', d)\widehat{q}(y'', d)}
\end{aligned}
$$

**Algorithm 4** DDFA (Naïve) Prediction

---

**input** $\widehat{\mathbf{Q}}_{c(X)|Y}, \widehat{\mathbf{Q}}_{Y|D}$, clustering discretization function $c, (x', d') \sim q(x, d)$

1: Pass $x'$ through $c$ to get cluster id $c(x')$.
2: $\widehat{q}(c(x')|Y = y'') \leftarrow [\widehat{\mathbf{Q}}_{c(X)|Y}]_{c(x'),y''}$ for all $y''$
3: $\widehat{q}(y''|D = d') \leftarrow [\widehat{\mathbf{Q}}_{Y|D}]_{y'',d'}$ for all $y''$
4: $\widehat{q}(y|c(X) = c(x'), D = d') \leftarrow \dfrac{\widehat{q}(c(x')|Y = y)\widehat{q}(y|D = d')}{\sum\limits_{y'' \in \mathcal{Y}} \widehat{q}(c(x')|Y = y'')\widehat{q}(y''|D = d')}$

5: $y_{\text{pred}} \leftarrow \arg\max_{y \in [k]} \widehat{q}(y|c(X) = c(x'), D = d')$
**output** : $\widehat{q}(y|c(X) = c(x'), D = d') = \widehat{p}_{d'}(y|c(x')), y_{\text{pred}}$

---

By label shift, $q(c(x)|y, d) = q(c(x)|y)$, then

$$
\begin{aligned}
\widehat{p}_d(y|c(x)) = \widehat{q}(y|d, c(x)) &= \frac{\widehat{q}(c(x)|y)\widehat{q}(y, d)}{\sum\limits_{y'' \in \mathcal{Y}} \widehat{q}(c(x)|y'')\widehat{q}(y'', d)} \\
&= \frac{\widehat{q}(c(x)|y)\widehat{q}(y|d)\widehat{q}(d)}{\sum\limits_{y'' \in \mathcal{Y}} \widehat{q}(c(x)|y'')\widehat{q}(y''|d)\widehat{q}(d)} \\
&= \frac{\widehat{q}(c(x)|y)\widehat{q}(y|d)(1/r)}{\sum\limits_{y'' \in \mathcal{Y}} \widehat{q}(c(x)|y'')\widehat{q}(y''|d)(1/r)} \\
&= \frac{\widehat{q}(c(x)|y)\widehat{q}(y|d)}{\sum\limits_{y'' \in \mathcal{Y}} \widehat{q}(c(x)|y'')\widehat{q}(y''|d)}
\end{aligned}
$$

Combining Algorithms 3 and 4 allows us to empirically evaluate the behavior of an ablation on DDFA which does not use any domain discriminator. For a reasonable comparison, we need a meaningful naïve representation space. We use a SCAN pretrain backbone for ResNet-18, and remove the last linear layer in the ResNet-18 backbone in order to expose a 512-dimension representation space. Since clustering in high-dimensional spaces often performs poorly, we also map this 512-dimension representation down to only $r$ (the number of domains) dimensions using two different common dimensionality reduction methods: Independent Component Analysis (ICA) [40] and Principal Component Analysis (PCA) [75]. These smaller-dimension clustering problems provide a closer comparison to the dimensionality of the clustering problem in the DDFA (SI) procedure, for which we employ $m$ clusters. Note: we use ICA and PCA implementations from scikit-learn [59].

SCAN, DDFA (RI), and DDFA (SI) experiment details are the same as explained in App. D; in fact, these are the same trials as in Sec. 6.

In general, we can see that the Naïvely ablated DDFA procedure performs worse than DDFA (SI) approaches in all problem settings, over both metrics of interest. However, it usually outperforms DDFA (RI). The ICA and PCA variants of the Naïve ablation generally underperform the Naïve ablation.

Table 7: *Extended Results on CIFAR-20.* Each entry is produced with the averaged result of 3 different random seeds. With DDFA (RI) we refer to DDFA with randomly initialized backbone. With DDFA (SI) we refer to DDFA's backbone initialized with SCAN. Note that in DDFA (SI) and DD (SI), we do not leverage SCAN for clustering. With Naïve we refer to an ablation in which DDFA's domain discriminator is replaced with the SCAN pretrained backbone, with its final linear layer removed so that its output is a 512-dimension unsupervised representation space. With Naïve (ICA) and Naïve (PCA) we refer to similar ablations in which the activations from the second-to-last layer of SCAN network are mapped to $r$-dimensional space with ICA and PCA respectively. $\alpha$ is the Dirichlet parameter used for generating label marginals in each domain, $\kappa$ is the maximum allowed condition number of the generated $\mathbf{Q}_{Y|D}$ matrix, $r$ is number of domains.

| r | Approaches | $\alpha : 0.5,\ \kappa : 8$ | | $\alpha : 3,\ \kappa : 12$ | | $\alpha : 10,\ \kappa : 20$ | |
|---|---|---|---|---|---|---|---|
| | | Test acc | $\mathbf{Q}_{Y|D}$ err | Test acc | $\mathbf{Q}_{Y|D}$ err | Test acc | $\mathbf{Q}_{Y|D}$ err |
| 20 | SCAN | $0.454 \pm 0.016$ | $0.059 \pm 0.002$ | $0.421 \pm 0.010$ | $0.051 \pm 0.001$ | $\mathbf{0.436} \pm 0.009$ | $0.038 \pm 0.001$ |
| | DDFA (RI) | $0.520 \pm 0.064$ | $0.041 \pm 0.005$ | $0.357 \pm 0.020$ | $0.043 \pm 0.003$ | $0.187 \pm 0.008$ | $0.051 \pm 0.003$ |
| | DDFA (SI) | $\mathbf{0.852} \pm 0.015$ | $\mathbf{0.015} \pm 0.000$ | $\mathbf{0.548} \pm 0.062$ | $\mathbf{0.026} \pm 0.003$ | $0.354 \pm 0.096$ | $\mathbf{0.036} \pm 0.007$ |
| | Naïve | $0.594 \pm 0.018$ | $0.045 \pm 0.004$ | $0.417 \pm 0.034$ | $0.047 \pm 0.003$ | $0.311 \pm 0.011$ | $0.045 \pm 0.003$ |
| | Naïve (ICA) | $0.311 \pm 0.025$ | $0.073 \pm 0.005$ | $0.221 \pm 0.003$ | $0.068 \pm 0.006$ | $0.190 \pm 0.012$ | $0.060 \pm 0.001$ |
| | Naïve (PCA) | $0.402 \pm 0.007$ | $0.060 \pm 0.003$ | $0.279 \pm 0.013$ | $0.056 \pm 0.006$ | $0.184 \pm 0.016$ | $0.052 \pm 0.004$ |
| 25 | SCAN | $0.458 \pm 0.055$ | $0.059 \pm 0.004$ | $0.455 \pm 0.011$ | $0.048 \pm 0.001$ | $0.440 \pm 0.023$ | $0.037 \pm 0.001$ |
| | DDFA (RI) | $0.525 \pm 0.033$ | $0.042 \pm 0.006$ | $0.310 \pm 0.020$ | $0.051 \pm 0.006$ | $0.182 \pm 0.002$ | $0.053 \pm 0.003$ |
| | DDFA (SI) | $\mathbf{0.819} \pm 0.013$ | $\mathbf{0.021} \pm 0.002$ | $\mathbf{0.707} \pm 0.022$ | $\mathbf{0.022} \pm 0.004$ | $\mathbf{0.502} \pm 0.024$ | $\mathbf{0.030} \pm 0.002$ |
| | Naïve | $0.547 \pm 0.074$ | $0.049 \pm 0.007$ | $0.457 \pm 0.035$ | $0.043 \pm 0.002$ | $0.324 \pm 0.038$ | $0.040 \pm 0.004$ |
| | Naïve (ICA) | $0.262 \pm 0.021$ | $0.078 \pm 0.003$ | $0.214 \pm 0.028$ | $0.064 \pm 0.003$ | $0.183 \pm 0.009$ | $0.061 \pm 0.000$ |
| | Naïve (PCA) | $0.356 \pm 0.019$ | $0.061 \pm 0.005$ | $0.216 \pm 0.020$ | $0.061 \pm 0.005$ | $0.189 \pm 0.008$ | $0.053 \pm 0.003$ |
| 30 | SCAN | $0.456 \pm 0.012$ | $0.059 \pm 0.001$ | $0.441 \pm 0.023$ | $0.050 \pm 0.001$ | $0.437 \pm 0.010$ | $0.037 \pm 0.001$ |
| | DDFA (RI) | $0.506 \pm 0.104$ | $0.045 \pm 0.009$ | $0.256 \pm 0.007$ | $0.058 \pm 0.005$ | $0.088 \pm 0.016$ | $0.076 \pm 0.006$ |
| | DDFA (SI) | $\mathbf{0.845} \pm 0.041$ | $\mathbf{0.020} \pm 0.006$ | $\mathbf{0.688} \pm 0.023$ | $\mathbf{0.027} \pm 0.003$ | $\mathbf{0.531} \pm 0.034$ | $\mathbf{0.029} \pm 0.002$ |
| | Naïve | $0.512 \pm 0.015$ | $0.052 \pm 0.004$ | $0.453 \pm 0.010$ | $0.043 \pm 0.002$ | $0.338 \pm 0.012$ | $0.038 \pm 0.001$ |
| | Naïve (ICA) | $0.280 \pm 0.065$ | $0.074 \pm 0.008$ | $0.183 \pm 0.013$ | $0.071 \pm 0.004$ | $0.146 \pm 0.018$ | $0.062 \pm 0.004$ |
| | Naïve (PCA) | $0.383 \pm 0.020$ | $0.059 \pm 0.004$ | $0.254 \pm 0.027$ | $0.059 \pm 0.008$ | $0.186 \pm 0.022$ | $0.048 \pm 0.003$ |