# OpenReview forum: "Unsupervised Learning under Latent Label Shift"
_NeurIPS.cc/2022/Conference — NeurIPS 2022 Accept_

### Official Review · Reviewer_3GNB · 2022-07-07

**Rating:** 7
**Confidence:** 4
**Soundness:** 3 good
**Presentation:** 4 excellent
**Contribution:** 3 good

**Summary:**

This work tackles the unsupervised learning problem under latent label shift. In this problem, each data point has a domain label, but the true classification label is not available. The key underlying assumption is that the class conditional distribution is domain invariant, while the distribution of labels in each domain affects the final data distribution. To resolve this problem, this paper relates it to the classical topic modeling problem and further extends its framework to the continuous domain. Empirically, the proposed method outperforms the SCAN method on CIFAR and FieldGuide datasets.

**Questions:**

1. Note that the 'pre-training backbone' step still relies on the feature-space similarity assumption. I doubt whether the main goal of this work that 'unsupervised learning can work beyond feature similarity' still holds? Or maybe similarity learning is still an essential step for unsupervised learning, but this work takes a further step on a coarser granularity? For example, we still have to find the sub-clusters of caterpillars and butterflies, but they belong to the same super-cluster.

2. How does DDFA (RI) perform on FieldGuide datasets? It's weird that different variants of DDFA are used on different datasets.

3. On the FieldGuide-2 dataset, the Q{Y|D} error seems to be high, why? Why is the final performance not affected by a bad estimation of Q_{Y|D}?

4. Are the limitations of this work described anywhere? I notice that the authors answered Yes in the checklist.

**Strengths And Weaknesses:**

Pros:

1. While current unsupervised learning techniques are mostly driven by the similarity assumption among data points, this work helps understand the conditions in which unsupervised learning is possible. I believe this work brings new insights to the community.
2. The linkage between the LLS problem with the topic modeling is intuitive and convincing.
3. This work is original and well-written.

Cons:

1. My main concern lies in the clustering step in their algorithm, which is mainly designed to discretize the input such that they act like words. This can be problematic in the high-dimensional case, where the identifiability assumption can also fail. I suppose this is the reason that DDFA relies on a well-trained backbone in experiments.
2. The experiments are sort of weak; see my question below.

---

> ### Author Response · Authors · 2022-08-02
> **Response to Reviewer 3GNB**
>
> Thank you for your constructive feedback on the experimentation, problem setting, and clarity of work; we have addressed your questions and concerns below and made improvements to the work accordingly.
>
> __“My main concern lies in the clustering step in their algorithm, which is mainly designed to discretize the input such that they act like words. This can be problematic in the high-dimensional case, where the identifiability assumption can also fail.”__
>
> Indeed, the clustering procedure is fallible in high dimensions, although we mention that clustering in (# of domains) space is often more reasonable than input space or some larger feature space. In general, we regard the clustering procedure as a heuristic procedure which invokes a tradeoff between tolerance to noise in $q(d|x)$ prediction and potential degeneracy in high dimensions.
>
> __“Q1. Note that the 'pre-training backbone' step still relies on the feature-space similarity assumption. I doubt whether the main goal of this work that 'unsupervised learning can work beyond feature similarity' still holds? Or maybe similarity learning is still an essential step for unsupervised learning, but this work takes a further step on a coarser granularity?”__
>
> In fact, it is true that our approach relies on similarity in the space induced by predicting $q(d|x)$. We regard the key difference between this approach and a naive approach to be the semantic meaning of the similarity. In an arbitrary feature space, perhaps in a learned self-supervised neural network, similarity has no well-defined meaning, while in a learned discriminator space, it is an approximation of the probability $q(d|x)$, meaning that if the learned discriminator achieved a minimal global loss, similarity in this space directly corresponds to class membership.
>
> We do find that the unsupervised pre-training approaches with which we initialize the backbone are in fact empirically able to recover useful features (if not, SCAN could not achieve its remarkably high accuracies). In some sense, our approach is able to leverage those quality unlabeled features in a well-structured way in order to produce accurate $q(d|x)$ predictions.
>
> Your argument about sub-clusters is a good way to reason about the benefits to be found when combining a well-defined learning objective like domain discrimination with high-quality pretraining–although it might require further analysis to see if sub-clusters are really induced in any pretrain feature space. We might instead say, more broadly, that the pre-train steps learn useful representations, and then the domain discrimination task outputs a structured representation that allows us to merge together similar sub-clusters when producing final class boundaries.
>
> __“Q2. How does DDFA (RI) perform on FieldGuide datasets?”__
>
> We have added FieldGuide-2 results with random initialization to the revision. We were unable to add FieldGuide-28 or ImageNet results with random initialization for this revision, but would target these additions for any camera-ready revision. In general, the reason for omission was purely that training these models is computationally intensive, as they involve many epochs of training on ResNet50 backbones with larger image input sizes than CIFAR, and we must evaluate across many different problem setups. Furthermore, results on CIFAR suggested that SCAN + DDFA nearly always surpassed DDFA alone, so we focused on the approach we expected to be best.
>
> **“Q3. On the FieldGuide-2 dataset, the Q{Y|D} error seems to be high, why? Why is the final performance not affected by a bad estimation of Q_{Y|D}?”**
>
> A simple explanation for why average per-entry error is high for FieldGuide-2 is that these $Q_{Y|D}$ matrices have many fewer entries per column than the remainder of the datasets (2, versus 10, 20, 28, or 50). Since columns always sum to 1, the average entry is now 0.5, versus 0.1, 0.05, and so on. Then variation in the entries is more pronounced. The final performance may not be negatively affected because of the argmax effect: prediction accuracy is based on top-1 probability in the $p_d(y|x)$ vector, and so even if this probability is miscalibrated, as long as the right value is the largest, it will produce the right result.
>
> __“Q4. Are the limitations of this work described anywhere?”__
>
> We acknowledge that a condensed summary of all limitations was indeed missing (and instead, our previous reference to limitations was more implicit–for example, our assumptions A.1-A.4 can be viewed as limitations as they describe a limited set of circumstances under which identification is guaranteed). We have added additional discussion of limitations in two key locations: 1) in the Discretize subsection in Section 5, we added language to describe the drawbacks of the clustering procedure, and 2) in Appendix A, we added a summary of limitations.

---

> > ### Comment · Reviewer_3GNB · 2022-08-04
> > **Post Rebuttal**
> >
> > I'd like to thank the authors for the informative response. My concerns have been properly addressed, and I would like to keep my original score.

---

### Official Review · Reviewer_GfX1 · 2022-07-09

**Rating:** 7
**Confidence:** 3
**Soundness:** 3 good
**Presentation:** 3 good
**Contribution:** 3 good

**Summary:**

Traditional unsupervised learning methods involve clustering according to similarity in some representation space. However, the true labels may not align with such clusters, for instance, if the clusters learn a spurious feature instead (such as adult versus caterpillar for moth versus butterfly classification). Instead, this paper proposes an unsupervised learning technique based on label shifted domains; that is, given unlabeled datasets coming from different domains where $p(x|y)$ stays the same (e.g. the within-class structure) but $p(y)$ changes, such as across time and location, we can uniquely recover the labels of points. First, the paper considers the tabular setting and draws an equivalence to topic modeling, where points are words, domains are documents, and labels are topics. A simple set of assumptions is presented for recoverability through non-negative matrix factorization. The next challenge is extending such an approach to continuous data. This is done by clustering examples based on the vectors $[p(d|x)]_d$ approximated by a classifier over the domains, performing the topic modeling approach on the discretization, and converting recovered quantities back to $p(y|d, x)$ using Bayes rule and the classifier. They show that on real-world datasets where feature space similarity does not imply the same label, this approach is able to outperform other unsupervised learning methods given synthetically induced label shifts.


**Questions:**

**Q1.** Could this approach be applied to the Waterbirds dataset (https://github.com/kohpangwei/group_DRO)? Would we expect it to do well? This is similar to FieldGuide, and I imagine that when naive clustering is done, things will be clustered by the birds’ background, which tends to dominate the image. I’m also curious what the effect of subgroup imbalance is here, since there are few land birds in water and few water birds on land. Equivalently, in FieldGuide, are there roughly the same amount of immature and adult moths/butterflies? Is your approach robust to these imbalances?

**Q2.** The importance of discretizing with f(x) could be made clearer by performing an ablation on some naive representation space.

**Q3.** Presentation-wise, it is not clear to me how theorem 2 and DDFA are related. I believe the connection is that asymptotically, DDFA satisfies theorem 2, but this is not explicit enough. Furthermore, a quick glance at Theorem 2 does not highlight how the identifiability is dependent on the discretization approach; it currently sounds like something that is just simply true.

**Q4.** Alpha (line 325) should be defined in the text. There is a typo in algorithm 1, line 8: return $\hat{\textbf{Q}}_{c(X) | Y}$? But that is discarded, right?


**Strengths And Weaknesses:**

**Strengths**

_Originality_: I found this paper to be very novel in establishing a connection between label shift and topic modeling.

_Quality_: theoretical and empirical results were solid. The theory felt straightforward, drawing directly from topic modeling literature and applying Bayes rule, but I think it was sufficient for the scope of this paper.

_Presentation_: the paper was well-written and easy to follow.

_Significance_: this method seems to be potentially very useful for handling datasets with spurious features that naive clusters would overindex on, which is a pervasive challenge in areas like medical imaging (Oakden-Rayner et. al., 2019).

Oakden-Rayner, L. et. al. Hidden Stratification Causes Clinically Meaningful Failures in Machine Learning for Medical Imaging. ML4C at NeurIPS, 2019.

**Weaknesses**

_Quality_: there were no direct theoretical results on the DDFA method. It would have been nice to discuss further tradeoffs for this discretization, such as choice of m and estimation error.

_Significance_: I am not sure how practically relevant this sort of approach can be. My first concern is in general about label shift, since I am not aware of if there are real datasets that cleanly capture this phenomenon. Second, based on the recoverability assumptions, if we had a task with many classes, we would need just as many domains/sources of data, such as those across time and location.

---

> ### Author Response · Authors · 2022-08-02
> **Response to Reviewer GfX1, Part 1**
>
> Thank you for your detailed feedback and examination of the empirical and theoretical aspects of the work. We appreciate the highlight of medical imaging as a potential area of high-impact for well-structured unsupervised learning.
>
> __“there were no direct theoretical results on the DDFA method. It would have been nice to discuss further tradeoffs for this discretization, such as choice of m and estimation error.”__
>
> We agree that it would be desirable to develop statistical theory, e.g. classifier error bounds and parameter estimation error bounds for DDFA in the future. However we believe that our initial set of contributions, including the identification theory and the well-motivated practical algorithm are sufficient for an initial contribution and are currently working on developing the statistical theory around the DDFA (and related) method(s).
>
> As for the choice of m, we have added a new empirical ablation study to Appendix H investigating how varying m can adjust performance of DDFA (SI) on CIFAR-20.
>
> __“My first concern is in general about label shift, since I am not aware of if there are real datasets that cleanly capture this phenomenon.”__
>
> We would like to open by commenting broadly on the role of theory and of stylized assumptions in distribution shift work. Absent any assumptions, the problems are fundamentally ill-posed. No method can work for all cases that we care about. Thus, research tends to fracture into different sides – some that focus on what seems to work in the cases commonly encountered in applications of interest, and others that probe what sorts of structures might render these problems tractable. Just as an architect must understand circles even if we seldom find perfect circles in nature or human structures, structural models of shift serve as idealized cartoons that help us to understand under what sorts of structures we can hope to make headway. After working out the case, our next steps are to relax the theory to make the mildest assumptions possible, and to identify real-world scenarios that are close (if not completely faithful).
>
> Analyzing in-depth the extent to which various existing datasets demonstrate label shift is an excellent topic for work. We stress that a healthy line of literature in this field has developed theoretical insights under the same label shift conditions we apply here, for example (Lipton et al. 2018) [4], (Azizzadenesheli et al. 2019) [1], or other works mentioned in our related works section. A common example in these papers concerns shifts in the prevalence of a disease where the constellation of symptoms (given the disease) does not shift.
>
> We remark that stringent theoretical assumptions are a useful sub-case that can pave the way for results using more relaxed assumptions. For example, in the identifiability literature for discrete topic modeling, (Donoho & Stodden 2003) [2] presented the anchor word condition for identifiability, and (Huang et al. 2016) [3] relaxed this condition to a broader condition known as “sufficiently scattered.” In a similar vein, we invite future work which tolerates some deviation from the domain-invariance of p(x|y), and therefore softens the domain conditions.
>
> [1] Kamyar Azizzadenesheli, Anqi Liu, Fanny Yang, and Animashree Anandkumar. Regularized learning for domain adaptation under label shifts. In International Conference on Learning Representations (ICLR), 2019.
>
> [1] David Donoho and Victoria Stodden. When does non-negative matrix factorization give a correct decomposition into parts? Advances in Neural Information Processing Systems (NeurIPS), 16, 2003.
>
> [2] Kejun Huang, Xiao Fu, and Nikolaos D Sidiropoulos. Anchor-free correlated topic modeling: Identifiability and algorithm. In Advances in Neural Information Processing Systems (NeurIPS), 2016.
>
> [4] Zachary Lipton, Yu-Xiang Wang, and Alexander Smola. Detecting and correcting for label shift with black box predictors. In International Conference on Machine Learning (ICML). PMLR, 2018.

---

> > ### Author Response · Authors · 2022-08-02
> > **Response to Reviewer GfX1, Part 2**
> >
> > __“Second, based on the recoverability assumptions, if we had a task with many classes, we would need just as many domains/sources of data, such as those across time and location.”__
> >
> > This is certainly true, and your callout of time and location as two potential paradigms for obtaining many domains of data is spot-on. We agree that this is not often satisfied when considering domain shift with a single source and single target set, but we also point out that the rich domain information provides the advantage that DDFA can exploit; with only a few domains, there is much less information available.
> >
> > __“Q1. Could this approach be applied to the Waterbirds dataset?…”__
> >
> > This dataset would be an excellent example to use to further strengthen our empirical results. In the time available for rebuttal, we were unable to complete the lengthy SCAN pretraining procedure on the Waterbirds dataset (which is necessary to provide SCAN-initialized experimental results, and, more important, is necessary to provide a suitable baseline). We target this for any potential camera-ready revision.
> >
> > __“Q1. I’m also curious what the effect of subgroup imbalance is here…in FieldGuide, are there roughly the same amount of immature and adult moths/butterflies?”__
> >
> > We do not have labeling granularity to the sub-group level, to be able to quantitatively check this definitively. However, when working with the dataset, we observed that at least some classes were heavily skewed toward the “adult” subgroup, with many fewer “larval/caterpillar” photos. For example, for this rebuttal, we randomly sampled 30 photos without replacement from the class label 0, and manually counted. We found 23 butterfly pictures, and 7 caterpillar/larval pictures.
> >
> > __“Q2. The importance of discretizing with f(x) could be made clearer by performing an ablation on some naive representation space.”__
> >
> > We agree, and we investigated this direction early in our work. In general, a high-dimensional naive representation space is extremely unsuited for the K-means clustering algorithm we apply in $q(d|x)$ space, and these results may not be meaningful.
> >
> > __“Q3. Presentation-wise, it is not clear to me how theorem 2 and DDFA are related. I believe the connection is that asymptotically, DDFA satisfies theorem 2, but this is not explicit enough.”__
> >
> > Each of the steps in DDFA are inspired by a step in the constructive identification of $Q_{Y|D}$ and $p_d(y|x)$ which is used for the proof of Theorem 2. However, it is important to remark that, at this time, we present no argument that DDFA will converge to the identifiable solution. The main obstacle to a clear convergence proof is the K-means clustering procedure, a heuristic to approximate point mass recovery which carries no guarantees about recovering the true anchor words. In particular, in the case in which the anchor subdomains do not contain all of the mass (equivalently, there are some $x$ which could belong to more than one $y$), the arbitrary distribution of mass outside of the anchor subdomains makes it very difficult to reason about the behavior of K-means.
> >
> > __“Q4. Alpha (line 325) should be defined in the text. There is a typo in algorithm 1, line 8: return Q^c(X)|Y? But that is discarded, right?”__
> >
> > We have added additional language to define alpha. Our algorithm does not have any use for the estimate of $Q_{c(X)|Y}$, using only the estimate of $Q_{Y|D}$. Nevertheless, we mark it here purely to reflect that the NMF algorithm produces 2 matrices that are the decomposed parts; we note that it is not returned from the end of the algorithm or used in any subsequent step.

---

> > > ### Author Response · Authors · 2022-08-08
> > > **Follow-up Response to Reviewer GfX1: Naive Representation Ablation**
> > >
> > > We have made some additional improvements to address the following question:
> > >
> > > __“Q2. The importance of discretizing with f(x) could be made clearer by performing an ablation on some naive representation space.”__
> > >
> > > Given a bit of additional time, we were able to evaluate a naive representation ablation on CIFAR-20 using three different naive representation spaces:
> > > - The 512-dimension activation directly before the last linear layer of the SCAN backbone
> > > - The same 512-dimension SCAN activation, but mapped into only (# of domains) dimensions with ICA
> > > - The same 512-dimension SCAN activation, but mapped into only (# of domains) dimensions with PCA
> > >
> > > The lower-dimension mappings are used to produce a representation space with the same dimensionality as the domain discriminator output space, to reduce the negative effect of high-dimensional clustering on the ablation.
> > >
> > > Performance is poor on these ablations, likely due to two reasons:
> > > - The lack of semantic meaning to the clustering space.
> > > - The inability to use the $q(d|x)$ predictions to perform domain-adjusted classification as per Algorithm 2. The modified procedure which does not rely on the domain discriminator at all essentially can only predict labels at the coarse granularity of a cluster.
> > >
> > > Ablation results, as well as a more detailed description of exactly how DDFA must be modified to work without a domain discriminator, are available in Appendix I (Page 30) of a revision we have just uploaded.

---

> > ### Comment · Reviewer_GfX1 · 2022-08-09
> > **Thank you for your response**
> >
> > Thank you so much for your response! I agree that we can always have broader discussion about if label shift is realistic, but this work is nonetheless valuable and theoretically interesting. I appreciate the additional experiments on clustering in a naive representation space and the additional discussion on limitations. I have updated my score and recommend acceptance.

---

### Official Review · Reviewer_tzBv · 2022-07-10

**Rating:** 5
**Confidence:** 4
**Soundness:** 4 excellent
**Presentation:** 4 excellent
**Contribution:** 4 excellent

**Summary:**

The authors introduce a novel general framework, referred to as Discriminate-Discretize-Factorize-Adjust (DDFA), for unsupervised learning under Latent Label Shift (LLS). With semi-syntheitc experiments, the authors show that the developed DDFA can leverage domain information to improve the performance of unsupervised classification.

**Questions:**

As far as I known, this work could be the first one to apply topic model to unsupervised learning under label shift. It does bring a lot of new ideas to me about how to make full use of topic models under multi-domain scenarios.

The whole work is built under the assumption of label shift, and all experiments are conducted with semi-syntheitc datasets. As the paper said, ``if we notice that whenever, ....we might conclude that the two were tied to the same underlying concept.’’, I absolutely agree with the point that topic model can be used to capture these co-occurrence patterns.

However, what if the assumption does not hold in the dataset. For instance, in the first domain, all dogs are under sun, but in the second domain, all dogs are under the moonlight. In that case, I assume that the topic models can hardly summarize these dogs into the same topic, and the developed DDFA will also fail.

So, I think the authors had better to prove whether the assumption really hold in real-world dataset, rather than use semi-syntheitc experiments.Then, it will be better to discover or visualize the underlying principle capture by DDFA, e.g, the amount of butterfly and caterpillar will shift together.

Additional Question

In Algorithm 2, it is quite confused to me that how you can obtain q(y|x) with Q(D|Y) and f(x). In my consideration, for a topic model, given a word x in the document d, it is unable to infer which topic the word assigned to.  So, which assumption can be used to infer the topic index y for each word x. You had better reexplained it under the scenario of topic model.


**Limitations:**

The novelty is high but the assumption is too idealistic.

**Strengths And Weaknesses:**

Strengths:

1) The principle ``instances that shift together group together’’ is interesting, and the authors creatively utilize topic model to capture these shits across various domains

2) The paper is well-organized and the proof is very rigorous.

Weakness:

1) My only concern is whether the label shift assumption exists in real-world datasets. Refer to the question part for details.

---

> ### Author Response · Authors · 2022-08-02
> **Response to Reviewer tzBv, Part 1**
>
> Thank you for your review. We are glad to see that you appreciated our theoretical results and particularly the novel connection to topic modeling.
>
> __“My only concern is whether the label shift assumption exists in real-world datasets…For instance, in the first domain, all dogs are under sun, but in the second domain, all dogs are under the moonlight…I think the authors had better to prove whether the assumption really hold in real-world dataset, rather than use semi-[synthetic] experiments.”__
>
> We would like to open by commenting broadly on the role of theory and of stylized assumptions in distribution shift work. Absent any assumptions, the problems are fundamentally ill-posed. No method can work for all cases that we care about. Thus, research tends to fracture into different sides – some that focus on what seems to work in the cases commonly encountered in applications of interest, and others that probe what sorts of structures might render these problems tractable. Just as an architect must understand circles even if we seldom find perfect circles in nature or human structures, structural models of shift serve as idealized cartoons that help us to understand under what sorts of structures we can hope to make headway. After working out the case, our next steps are to relax the theory to make the mildest assumptions possible, and to identify real-world scenarios that are close (if not completely faithful).
>
> We agree that the hypothetical situation you describe does not satisfy the label shift assumption (and therefore is poorly-suited for our algorithm). Analyzing in-depth the extent to which various existing datasets demonstrate label shift is an excellent topic for work. We stress that a healthy line of literature in this field has developed theoretical insights under the same label shift conditions we apply here, for example (Lipton et al. 2018) [4], (Azizzadenesheli et al. 2019) [1], or other works mentioned in our related works section. A common example in these papers concerns shifts in the prevalence of a disease where the constellation of symptoms (given the disease) does not shift.
>
> We remark that stringent theoretical assumptions are a useful sub-case that can pave the way for results using more relaxed assumptions. For example, in the identifiability literature for discrete topic modeling, (Donoho & Stodden 2003) [2] presented the anchor word condition for identifiability, and (Huang et al. 2016) [3] relaxed this condition to a broader condition known as “sufficiently scattered.” In a similar vein, we invite future work which tolerates some deviation from the domain-invariance of p(x|y), and therefore softens the domain conditions.
>
> [1] Kamyar Azizzadenesheli, Anqi Liu, Fanny Yang, and Animashree Anandkumar. Regularized learning for domain adaptation under label shifts. In International Conference on Learning Representations (ICLR), 2019.
>
> [1] David Donoho and Victoria Stodden. When does non-negative matrix factorization give a correct decomposition into parts? Advances in Neural Information Processing Systems (NeurIPS), 16, 2003.
>
> [2] Kejun Huang, Xiao Fu, and Nikolaos D Sidiropoulos. Anchor-free correlated topic modeling: Identifiability and algorithm. In Advances in Neural Information Processing Systems (NeurIPS), 2016.
>
> [4] Zachary Lipton, Yu-Xiang Wang, and Alexander Smola. Detecting and correcting for label shift with black box predictors. In International Conference on Machine Learning (ICML). PMLR, 2018.

---

> > ### Author Response · Authors · 2022-08-02
> > **Response to Reviewer tzBv, Part 2**
> >
> > __“In Algorithm 2, it is quite confused to me that how you can obtain q(y|x) with Q(D|Y) and f(x). In my consideration, for a topic model, given a word x in the document d, it is unable to infer which topic the word assigned to. So, which assumption can be used to infer the topic index y for each word x. You had better reexplained it under the scenario of topic model.”__
> >
> > We would add that it is important to recall that the computed $Q_{D|Y}$ matrix is subject to the same permutation of columns as the permutation of rows of $Q_{Y|D}$, which is factored into our identification of labels y only up to permutation. This is common in the topic modeling literature (indeed, the indexing of topics is arbitrary, and can be swapped arbitrarily without altering the quality of the solution), and for unsupervised classification (e.g. cluster 1 could be swapped in name with cluster 2). The only important thing is that the ordering of labels is consistent among all datapoints, so that class 1 consistently refers to the same underlying latent class.
> >
> > The proof of Lemma 1, which inspires the plug-in procedure of Algorithm 2, shows how the over-determined linear system can be solved to estimate $q(y|x)$ with the same ordering of classes y as in $Q_{D|Y}$. To reiterate a rough intuition of the procedure here: We begin by showing that $q(d|x)$ can be expressed as a $k$ term linear combination $\sum_{y=1}^{y=k} q(d|y)q(y|x)$. With access to $q(d|y)$, and $q(d|x)$ (from the domain discriminator), we note that this linear combination becomes a linear equation in $k$-unknowns. Since this is true for all domains in $[r]$, each domain offers one linear equation in $k$ unknowns. With our assumption of $r \geq k$, this becomes an overdetermined system and lends itself to simple solving. When the above quantities are handled in matrix form, with appropriate rank assumptions, solving the system corresponds to using the pseudo inverse that is mentioned in our algorithm. This pseudo inverse computes an estimate of the unknown quantity when the quantities we assume known are subject to noise.

---

### Official Review · Reviewer_RW2N · 2022-07-26

**Rating:** 7
**Confidence:** 4
**Soundness:** 3 good
**Presentation:** 4 excellent
**Contribution:** 3 good

**Summary:**

This paper introduces a new experimental setting $Unsupervised\ Latent\ Label\ Shift$, where the data are available from multiple domains and the label distributions vary across different domains but the class conditionals do not. Furthermore, this paper proposes a DDFA framework to solve the problem unsupervisedly and theoretically demonstrates that the matrix factorization solution may be identified. The experimental results demonstrate the effectiveness of the proposed framework.

**Questions:**

Please refer to the above section

**Limitations:**

Please refer to the above section

**Strengths And Weaknesses:**

Strengths:
1. The writing is very clear and the paper is easy to follow.
2. The setting of this paper is very interesting and practical.
3.  I appreciate the notion of using matrix factorization to calculate the distribution $Q_{X|D}$, which can save the cost on training the generative model $Q_{X|Y}$, making this framework realistic and easy to use in real-world applications.
4.More crucially, this work presents the assumptions and theoretic analysis of the $Q_{X|D}$ identifiability.


Weakness:
1. In the section "Discretize", the number of clusters $m$ is larger than $k$. Can the authors perform an ablation study on the ratio between $m$ and $k$? If so, the results may be more convincing.







It would be nice to have a citation for the relevant paper [1].

[1] Guo J, Gong M, Liu T, Zhang K, Tao D. Ltf: A label transformation framework for correcting label shift. International Conference on Machine Learning 2020 Nov 21 (pp. 3843-3853). PMLR.

---

> ### Author Response · Authors · 2022-08-02
> **Response to Reviewer RW2N**
>
> Thank you for your review. We appreciate your positive response to the theoretical identifiability and overall setting of the work.
>
> __“In the section "Discretize", the number of clusters m is larger than k. Can the authors perform an ablation study on the ratio between m and k?”__
>
> This ablation is added in Appendix H of the revision. On the DDFA (SI) model, on CIFAR-20, and with 20 domains, we see that performance is best when m is equal to or slightly larger than k; decreasing it below k or making it grow very large both decrease performance.
>
> __“It would be nice to have a citation for the relevant paper [1]. [1] Guo J, Gong M, Liu T, Zhang K, Tao D. Ltf: A label transformation framework for correcting label shift. International Conference on Machine Learning 2020 Nov 21 (pp. 3843-3853). PMLR.”__
>
> We recognize the relevance of the work you recommended, and upon review of its content, we agree that it is relevant when establishing the current situation of the field of label shift. We have acknowledged this work in our updated related work.

---

### Author Response · Authors · 2022-08-02
**General Response**

We would like to thank the reviewers for their feedback and comments. We appreciate that reviewers have recognized the paper to be original (tzBv, GfX1, 3GNB), the theoretical results to be “very rigorous” (tzBv) and “solid” (GfX1), and the setting to be “very interesting and practical” (RW2N). The connection between label shift and topic modeling was regarded as “very novel” (GfX1), “intuitive and convincing” (3GNB), and “creative” (tzBv). We are also glad to see that reviewers appreciated the clarity of our presentation (RW2N, tzBv, GfX1, 3GNB).

The reviewers also brought up several constructive points about the problem setting and empirical results that we believe are addressable in the camera-ready version. We address each reviewer’s concerns in turn in the respective threads.

Reflecting the promised improvements, we have posted a revised version of our paper, marking all updates to the work with maroon highlighting (except in table entries).  A brief summary of key changes follows:

*Experiments*

- DDFA (RI) results, from three random seeds, are now included for FieldGuide-2. DDFA (RI) results for FieldGuide-28 and ImageNet are not available in this revision, but are targeted for any potential camera-ready release.
- A new ablation study, with plots, in Appendix H in which we examine the effect of the choice of number of clusters m.
Naive SCAN $Q_{Y|D}$ error estimates are now available for all experiments.
- Results for FieldGuide-2 and FieldGuide-28 available in the original main paper have been replaced in order to correct a mismatch between evaluation transforms which incorrectly disadvantaged our approach. The replacement results are the same as those originally available in the appendices submitted as part of our supplementary material.

*Clarity*

- The FieldGuide-2 and FieldGuide-28 tables have been moved to the appendices to conserve space in the paper while other modifications have required additional space. Any potential camera-ready release will likely include one or both of these tables in the additional permissible page.
- A consolidated summary of limitations is available in Appendix A.
- Additional language in Section 5 clarifying the role of the clustering heuristic, and moved key theoretical language about the “unique” property of the anchor subdomains from the Appendices to the assumption A.4.

---

### Meta-Review · Area_Chair_DEG4 · 2022-08-25

**Recommendation:** Accept
**Confidence:** Certain

**Metareview:**

This paper studies the problem of identifying classes in unlabeled data. It considers a little-to-not-studied setting, where data is available exhibiting latent class proportion shift. This means that p(y) changes, but by assumption the conditional feature distributions p(x|y) do not. The paper proposes a principle that features that shift together should group together in order to identify the latent classes. In both the tabular and continuous data settings, the paper shows theoretically and in practice that this principle with the associated assumptions is sufficient to label the data in an unsupervised setting.

The reviewers all agreed that the paper makes a strong contribution to a less studied area, and that the idea of exploiting label shift to identify classes is a significant advancement in this area. In particular, they liked the connection between the proposed setting and topic modeling. This leads to an approach for handling continuous data by clustering. The reviewers also generally felt the paper is clear and well written.

Much of the discussion related to the realism of the proposed setting and the assumptions. The authors are encouraged to include the mains points from these discussions in the final version.

**Award:**

No

---

### Decision · Program_Chairs · 2022-09-14

Accept